# Translocation mechanism of xeroderma pigmentosum group D protein on single-stranded DNA and genetic disease etiology

Tanmoy Paul[1,2], Chunli Yan [1,2], Grant Derdeyn-Blackwell[1,2] & Ivaylo Ivanov [1,2] ✉

XPD is a key nucleotide excision repair (NER) protein whose function is vital for genome integrity. During NER, XPD serves as a 5′−3′ single-strand DNA translocase that enables lesion scanning and verification in genomic DNA. Yet, its translocation mechanism is incompletely understood. Here we use molecular simulations and chain-of-replicas path optimization methods to model the ATP-driven translocation mechanisms of XPD and its bacterial homolog DinG, revealing all on-path metastable intermediates and corresponding kinetic rates. We identify the XPD(DinG) global domain motions that modulate the strength of DNA association at the opposing ends of the DNA-binding groove. During the ATP hydrolysis cycle, alternating weak and strong interactions at two defined groove constrictions enable DNA reptation and forward displacement of the ATPase. Moreover, we show that DNA- or ATP-binding residues directly involved in translocation are hotspots for genetic disease mutations. Thus, our findings shed light on the etiology of XPD-associated genetic syndromes.

XPD belongs to the superfamily 2 (SF2) of helicases with 5′−3′ polarity[1]. As a motor subunit of the TFIIH complex[2,3], XPD is essential for nucleotide excision repair (NER)[4,5]—a critical genome maintenance pathway. NER is responsible for eliminating a diverse array of bulky or helix-distorting lesions from genomic DNA[6–11]. The effectiveness of NER depends on a two-step process of initial damage detection followed by lesion verification. NER's two sub-pathways, global genome NER (GG-NER) and transcription-coupled NER (TC-NER), differ only in the initial damage recognition step[12,13] but converge prior to lesion verification[14]. It is precisely at the lesion verification stage that the helicase activity of XPD is recruited[15–20]. Lesion verification is vital for the targeted processing of cytotoxic or carcinogenic lesions, while avoiding futile repair[21]. Predictably, inherited XPD mutations that selectively impair GG-NER, TC-NER or cause partial transcription defects give rise to severe human genetic diseases—xeroderma pigmentosum (XP), trichothiodystrophy (TTD), and Cockayne syndrome (CS)[18,22–26]. The phenotypes associated with these disorders are strikingly different. XP is characterized by UV-sensitivity and extreme cancer predisposition, while TTD and CS are characterized by accelerated neurodegeneration, recurring infections, and high mortality at a young age. The remarkable diversity in clinical outcomes is only partially understood in terms of structure and biological mechanisms.

XPD features two RecA-like core motor domains, RecA1 and RecA2, and two auxiliary domains, iron-sulfur (Fe−S) and Arch domains (Fig. 1a, b). The Fe−S, RecA1, and Arch domains come together to form a narrow channel that accommodates the passage of single-stranded DNA (ssDNA)[19,27,28]. The Fe−S domain harbors an iron-sulfur [4Fe−4S] cluster, which is necessary for the domain's structural integrity. The Arch domain is situated at the top of the central cleft between the core motor domains, contacting the Fe−S domain near the 3′-end of the bound ssDNA. XPD's DNA-binding groove spans the RecA1 domain and extends along the interface of the Arch and Fe−S domains. The interface of the two motor domains encompasses a highly conserved APT-binding site[1,28] (Fig. 1c) that features seven conserved helicase motifs

---

[1]Department of Chemistry, Georgia State University, Atlanta, Georgia, USA. [2]Center for Diagnostics and Therapeutics, Georgia State University, Atlanta, Georgia, USA. ✉e-mail: iivanov@gsu.edu

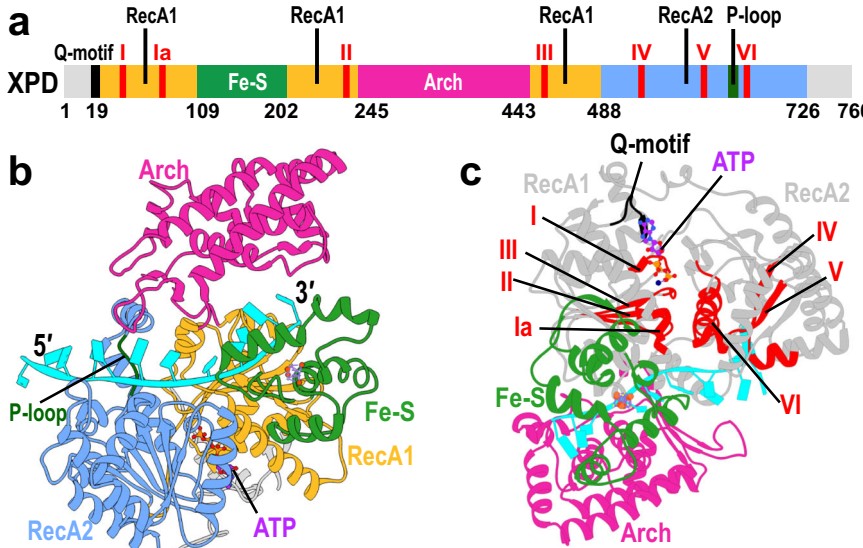

**Fig. 1 | Structural model of XPD with bound ATP and single-stranded DNA.**
**a** Schematic representation of XPD's domain organization along its sequence. Conserved helicase motifs (I, Ia, II–VI) are indicated in red. The less conserved Q-motif and P-loop are shown in black and dark green, respectively. **b** XPD structure shown in cartoon representation. RecA1 (gold), RecA2 (blue), Arch (magenta), and Fe–S (green) domains are shown, with the *N*-terminal and *C*-terminal ends of the protein chain colored in gray. Bound ssDNA is depicted in cyan with the 5′ and 3′ ends labeled. The ATP molecule bound at the RecA1–RecA2 interface is shown in atomic representation and colored by atom type. **c** Side view of the ATP-binding cleft, highlighting conserved motifs I, Ia, II–VI (red) and the Q-motif (black). ATP is shown in ball-and-stick representation and colored by atom type.

(motifs I, Ia, II–VI) (Supplementary Fig. 1, Supplementary Movie 1). The N-termini of RecA1 and RecA2 also feature the less conserved Q-motif and the P-loop. The width of the DNA groove changes reciprocally in response to DNA loading and ATP binding/hydrolysis—a key structural requirement for both XPD's translocase activity and NER.

The SF1A family of helicases provides a paradigmatic example of how DNA polarity is established in this class of proteins. SF1A helicases translocate in the 3′–5′ direction and engage in a cyclic process of alternating affinities[1,29]. ATP binding at the initiation of the translocation cycle prompts a rotational movement of the RecA1 domain toward RecA2, while RecA2 initially maintains a firm grip on the DNA. Subsequent formation of an ATP-stabilized interface between RecA1 and RecA2 alters the DNA-binding affinities of the motor domains, shifting the dominant interaction to RecA1. Ultimately, ATP hydrolysis and subsequent ADP release lead to the release of the RecA1 domain's hold on DNA and restore the helicase to its apo conformation. Thus, to attain reverse polarity, two mechanistic scenarios are conceivable: 1) reversing the orientation of the helicase scaffold with respect to ssDNA while maintaining a similar translocation mechanism; or 2) keeping the helicase scaffold orientation while reversing the order in which RecA1 and RecA2 bind tightly to ssDNA. The SF1A and SF1B sub-families (with 3′–5′ and 5′–3′ polarity, respectively) operate by the latter mechanism[29,30]. However, the mechanism governing translocation polarity in XPD and other SF2 family helicases is at present unknown.

XPD homologs are found across archaea, bacteria, and eukaryotes. Specifically, DinG is an XPD homolog from *Escherichia coli*[31] for which a simplified two-step translocation mechanism has been proposed[32]. In the first step, RecA2 moves along the ssDNA in the 5′–3′ direction, closing the ATP-binding cleft. The Arch domain concurrently rotates away from the Fe–S domain, opening the 3′-end of the DNA-binding cleft. In the second step, the Arch and RecA2 domains restore their initial orientations, thus pushing the ssDNA forward. Both XPD and DinG advance by one DNA base per ATP cycle. Given their considerable structural differences[32], it is not known if DinG and XPD share the same mechanism. Therefore, we modeled and compared the translocation mechanisms of the two translocases. To this end, we performed molecular dynamics simulations, partial nudged elastic band path optimizations, transition path sampling, and Markov State modeling on the two systems. From our analyses, we present a detailed molecular-level view of XPD and DinG translocation as a series of transitions among metastable states. Analysis of the metastable states yields insights into the conformational switching of XPD's domains during the different stages of the translocation process and their impact on ssDNA displacement. Moreover, we link the positioning and dynamics of XPD-associated disease mutations to key residues responsible for ssDNA translocation.

## Results

### XPD's translocation on DNA proceeds through seven distinct functional states

To investigate the translocation mechanism of XPD, we constructed molecular models of XPD–ssDNA complexes in apo-, ADP-, and ATP-bound states and explored their conformational transitions using partial nudged elastic band (PNEB) simulations and unbiased sampling along the computed transition paths. We then used time-lagged independent component analysis (TICA) to project the simulation-derived ensembles onto two-dimensional (2D) free energy surfaces (Fig. 2). Our analysis revealed multiple metastable states corresponding to distinct stages of the ATPase cycle (Fig. 2a). Markov state modeling of the combined trajectories identified ten macrostates, seven of which (S1–S7) form a sequential pathway that defines the DNA translocation mechanism (Fig. 2b). The remaining three macrostates represent kinetic traps. We then extracted frames from each macrostate and computed cluster centroids, yielding representative conformers of the seven functional states. We note that states S1 and S7 feature as separate minima on the projected free energy landscape. This observation reflects the fact that the TICA projection involves not only protein–protein but also protein–DNA distances. Thus, the S1 and S7 states differ only in terms of their ssDNA binding mode. To further clarify this point, we repeated the TICA and MSM analysis, excluding all protein-DNA contacts. The resulting free energy landscape (Supplementary Fig. 2a) only reflects the protein conformational space. We recover a closed PNEB path projected onto the 2D free energy surface as expected for a cyclic process of nucleotide binding and hydrolysis.

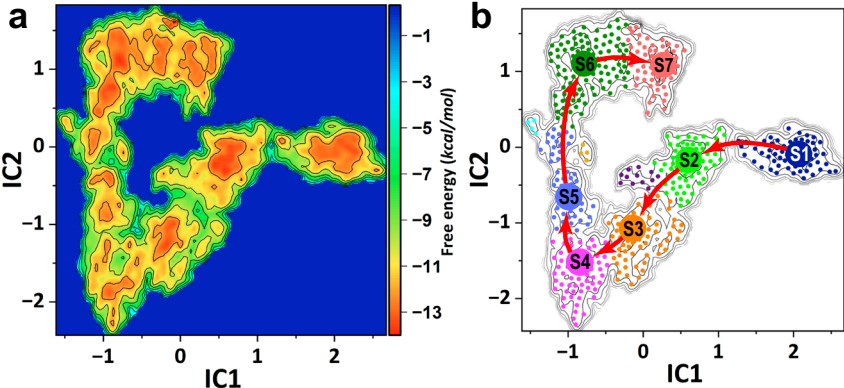

**Fig. 2 | Seven on-path conformational states define the DNA translocation mechanism of XPD. a** Free energy landscape underpinning XPD dynamics over a single ATPase cycle. The free energy is projected onto the first two ICs obtained from time-lagged independent component analysis. The color bar represents the free energy scale in kcal/mol. **b** Markov state model (MSM) built from the conformational ensemble sampled along the optimal transition path. Microstates (dots) are colored according to their macrostate assignment as determined by the PCCA+ algorithm. Seven macrostates traversed by the minimum free energy path (MFEP) are labeled. The remaining macrostates (purple, yellow, and cyan dots) correspond to kinetic traps and are not labeled.

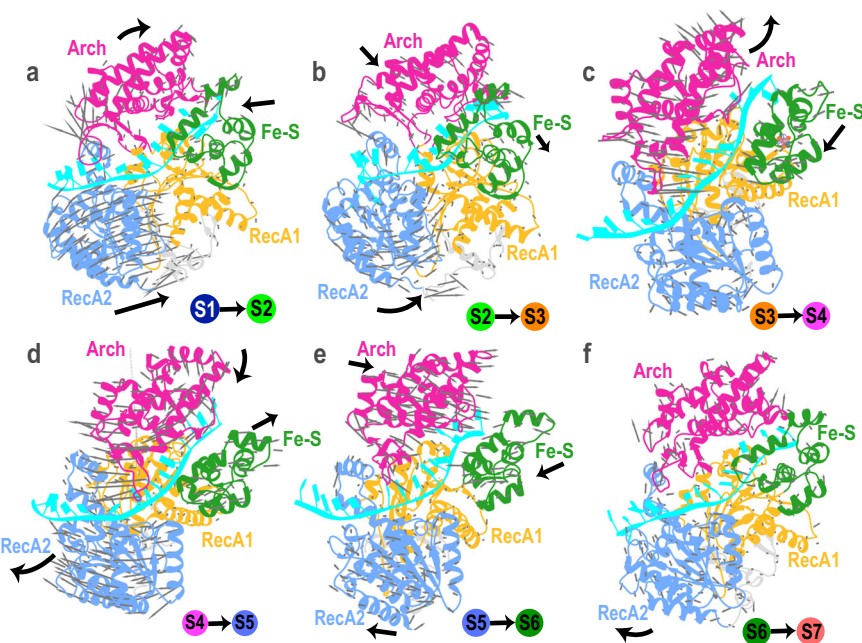

**Fig. 3 | Conformational switching of XPD's domains during the ATPase cycle.** Porcupine plots showing the principal conformational changes in XPD during a single cycle of ATP binding and hydrolysis. XPD domains are colored as follows: RecA1 (gold), RecA2 (blue), Arch (magenta), and Fe–S (green). Gray arrows indicate the direction of Cα atom displacements during structural transitions among the seven MSM macrostates (S1–S7): (**a**) S1 → S2; (**b**) S2 → S3; (**c**) S3 → S4; (**d**) S4 → S5; (**e**) S5 → S6; and (**f**) S6 → S7. Black arrows indicate the overall direction of large-scale domain movements. Terminal conformational states for each transition are indicated in the bottom right corner of each panel.

We also mapped representative conformers from states S1–S7 onto the new landscape. While the intermediate states (S3–S6) are well separated in the projected space defined by the protein coordinates, states S1, S2, and S7 appear as overlapping (Supplementary Fig. 2b). Distinguishing these states requires inclusion of the ssDNA coordinates.

## DNA translocation involves ATP-induced rotations of XPD's RecA2 and Arch domains

The optimal PNEB path uncovers the global motions of the XPD domains in response to ATP binding and hydrolysis (Supplementary Movie 2). The porcupine plots in Fig. 3 show the directions of atomic displacement leading to transitions from one functional state to the next for states S1–S7. Here, we present the analysis of just one of the

independently simulated ATPase cycles, with the other two yielding practically identical results (Supplementary Fig. 3). The cycle endpoints (states S1 and S7) both represent the apo state of XPD and differ only by the forward displacement of the ssDNA by one nucleotide. The intervening macrostates are all intermediates with S4 and S5 binding ATP and ADP, respectively. The largest domain rearrangements involve four states (S1, S4, S5, S7) and are highlighted in the subsequent discussion.

The three-step transition from S1 to S4 represents the overall closure of the ATPase cleft upon ATP association. In the first step (S1 → S2), the RecA2 domain senses the presence of a bound nucleotide and undergoes an inward shift toward RecA1, partially closing the gap between the two motor domains (Fig. 3a). In the subsequent S2 → S3

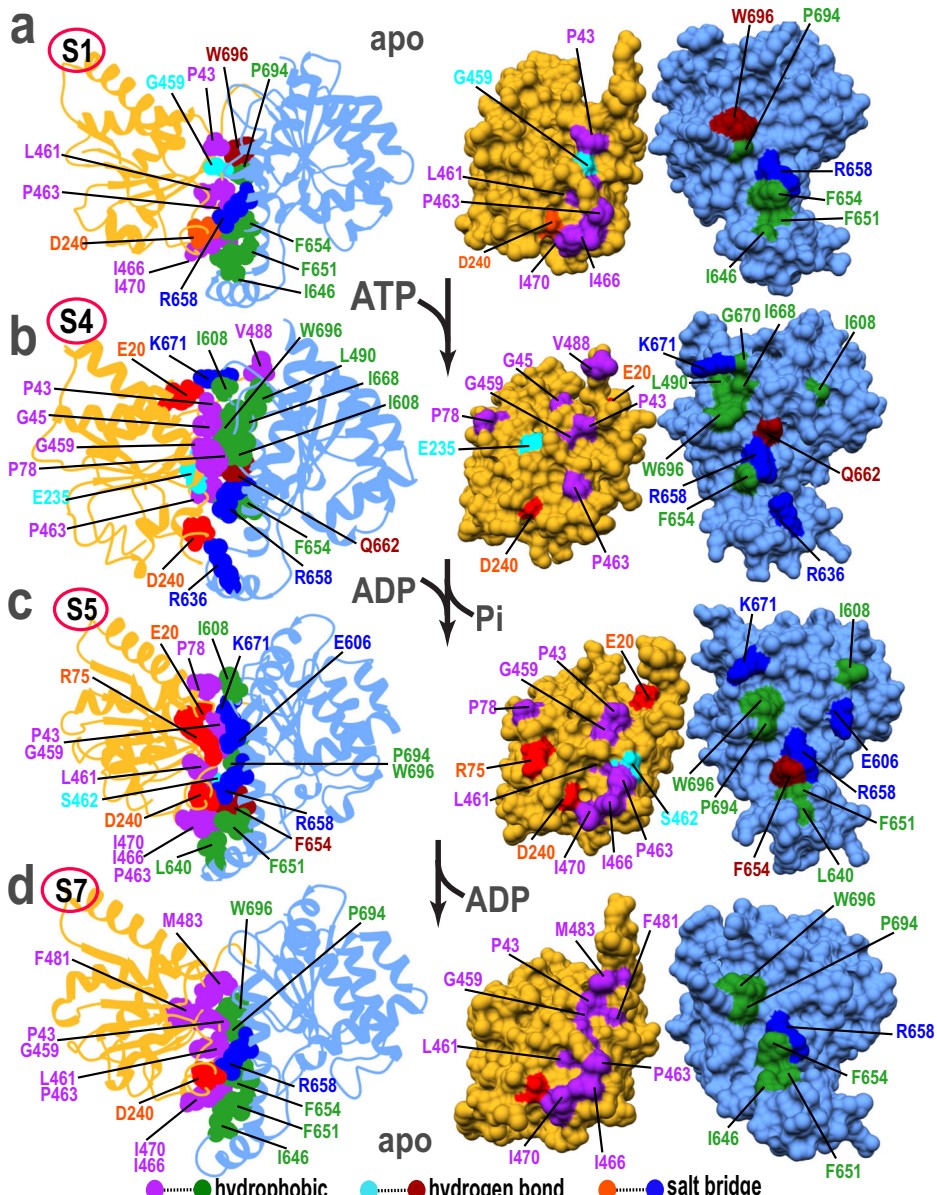

**Fig. 4 | Changes in persistent residue contacts at the RecA1–RecA2 interface during the ATPase cycle.** A zoomed-in view of the XPD ATPase cleft in states: (**a**) S1; (**b**) S4; (**c**) S5; and (**d**) S7. Panels on the left show XPD residues that are key to the structural integrity of the motor domains interface. Panels on the right show the same key contacts mapped onto the surfaces of the RecA1 (gold) and RecA2 (blue) domains in 'open-book' representation. Color-coding is by interaction type: 1) salt bridge-forming residues are in red and blue, 2) hydrogen bonding residues are in cyan and dark brown, and 3) residues involved in hydrophobic contacts are in purple and green.

transition, the RecA2 domain undergoes further conformational shift, resulting in progressive narrowing of the ATPase cleft (Fig. 3b). Subsequently, the ATP-induced complete closure of the RecA1–RecA2 cleft (S3 → S4) triggers a coupled rotation and reorientation of the XPD Arch domain (Fig. 3c) that opens the constricted 3'-end of the DNA-binding groove. This opening movement is accompanied by an increase in the center-of-mass distance $d_{A-F}$ between the Arch and Fe–S domains and a concomitant increase in the angle $\theta$ defined by the centers of mass of the Fe–S, RecA1, and Arch domains (Supplementary Fig. 4b, c). The full opening of the gap between the Fe–S and Arch domains in state S4 allows loading and unimpeded passage of ssDNA through the XPD DNA-binding cleft. By contrast, in the ADP-bound state S5, the gap recloses while the nucleotide binding site undergoes partial opening (Fig. 3d). Subsequent ADP dissociation induces a two-step (S5 → S6 and S6 → S7) conformational switch that fully opens the

ATP-binding cleft, restoring the apo conformation of the XPD helicase (Fig. 3e, f).

Next, we analyzed the XPD domain interfaces to identify persistent residue contacts in states S1–S7. Throughout this study, contacts were defined as persistent if detected in more than 75% of the aggregate trajectory frames from the MD ensemble. We further classified the contacts by type—hydrogen bonding, hydrophobic and salt bridge interactions. Results are presented in Fig. 4, Supplementary Fig. 5 and allow us to trace changes in interfacial contacts during conformational switching among the XPD functional states. Thus, we observe that the initial shift of RecA2 toward RecA1 during the S1 → S2 transition is driven primarily by long-range electrostatic interactions (Supplementary Fig. 6). Interactions between the two motor domains remain suboptimal in state S2 (Fig. 4a, Supplementary Fig. 5a) and a robust network of salt bridge interactions is not established until state S3

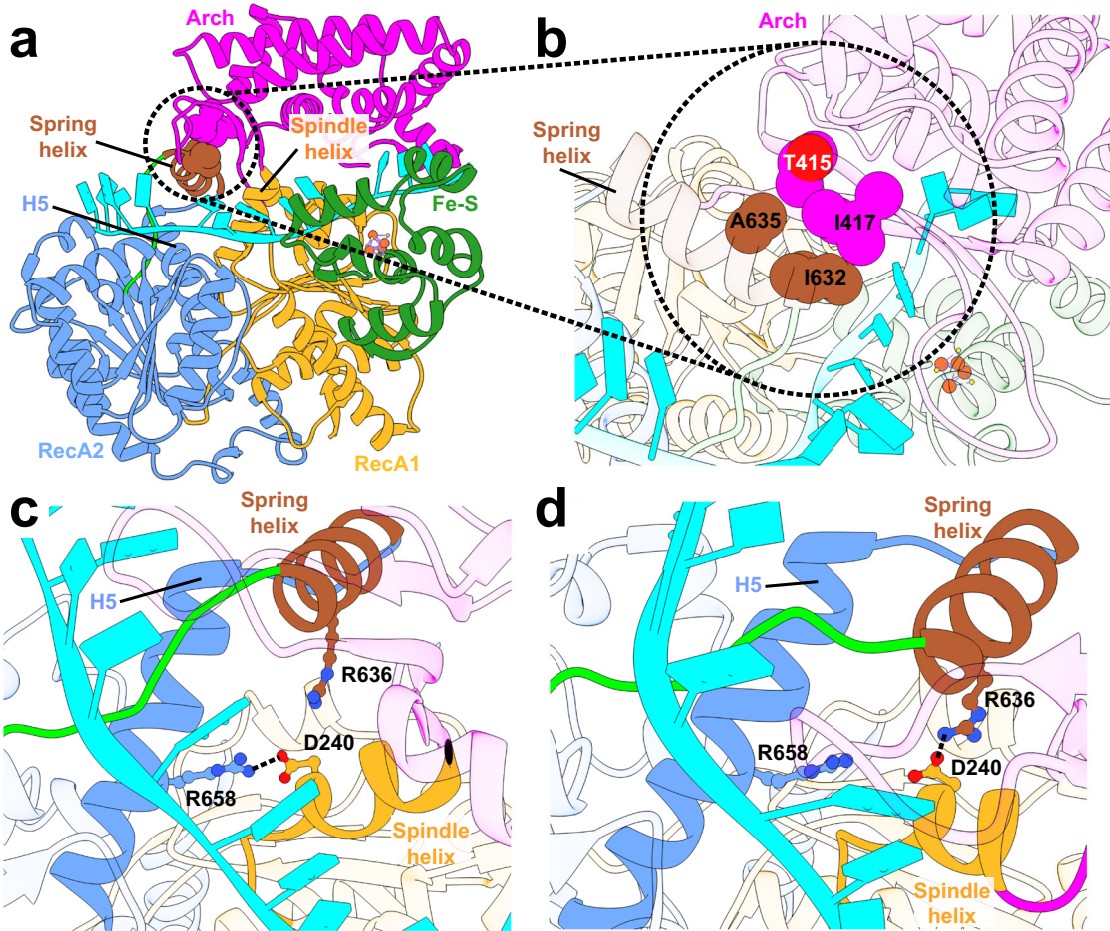

**Fig. 5 | Dynamic coupling of the RecA2 and Arch domains is mediated by key hydrophobic and salt-bridge interactions. a** A cluster of hydrophobic residues (highlighted by a circle) is positioned between the spring helix and the Arch domain and is key for allosteric communication from the motor domains to the Arch domain. The XPD spring helix is shown in brown; other domains are colored as follows: RecA1 (gold), RecA2 (blue), Arch (magenta), and Fe–S (green). Helix H5 and the spindle helix are labeled. **b** Zoomed-in view of the hydrophobic interface with interacting residues from the spring helix and the Arch domain shown as spheres and labeled. Salt-bridge interactions involving residues from helix H5, the spindle helix, and the spring helix in (**c**) the apo state, and (**d**) the ATP-bound state of XPD. Salt bridges are represented by black dashed lines.

(Supplementary Fig. 5b). In the S3→S4 transition, the helicase nucleotide binding pocket is reorganized to optimally accommodate ATP (Fig. 4b). The nucleotide serves as a molecular zipper closing the RecA1–RecA2 interface. Residues from XPD's conserved helicase motifs play a key role in this active site reorganization. Specifically, ATP's phosphate groups form: 1) hydrogen bonds with S44 and T49 (helicase motif I); 2) salt-bridges with K48 (helicase motif I), R666 and R669 (helicase motif VI); and 3) coordination bonds with $Mg^{2+}$ via D234 and E235 of helicase motif II. We also identified persistent hydrophobic interactions (π–π stacking) between the adenine base and the flanking aromatic residues Y14 and Y18 from the Q-motif. Additional hydrophobic contacts of ATP with F12, I17, and V50 were detected, which strengthen the RecA1–RecA2 interface. Beyond ATP-mediated interactions, other persistent contacts involved residues P78 (motif Ia) and I608 (motif V) (Fig. 4b). During the S4→S5 transition, ATP hydrolysis reduces the RecA1-RecA2 interdomain contacts resulting in partial opening of the nucleotide-binding cleft (Fig. 4c). ADP primarily engages with helicase motif I residues by forming hydrogen bonds with S44 and T49, and a salt bridge with K48. In the subsequent S5→S6 transition, ADP release disrupts the remaining interdomain contacts (Supplementary Fig. 5c), restoring the open apo conformation of the RecA1–RecA2 interface in S7 (Fig. 4d).

Notably, a helix from XPD's RecA2 domain, denoted as 'spring helix', was found to form persistent contacts with the Arch domain

(Fig. 5a, b) via a network of hydrophobic interactions (I632 and A635 from the spring helix contacting I417 and T415 of the Arch domain). By anchoring the Arch domain to RecA2, these contacts serve to convey the conformational changes triggered by ATP binding and hydrolysis from the motor domains interface to the distal regions of the Arch and Fe–S domains. Specifically, during the ATP binding phase (S1→S4), rotation of RecA2 generates mechanical torque that is propagated to the Arch domain, causing its outward displacement. Conversely, partial opening of the nucleotide-binding cleft after ATP hydrolysis causes retraction of RecA2, which in turn pulls back on the Arch domain via the spring helix, forcing the closure of the gap between the Fe–S and Arch domains in state S5. This reciprocal motion between the RecA2 and Arch domains, mediated by the identified hydrophobic network, underpins XPD's conformational switching in response to changing nucleotide states.

Our findings further reveal that the Arch domain undergoes rotation about a central 'spindle helix' (Fig. 5c, d), which is aligned with the ATPase cleft and oriented orthogonally to the direction of ssDNA propagation. The spindle helix contains an aspartate residue (D240), which plays a key role in the translocation mechanism. Two arginine residues originating from the spring helix (R636) and the immediately adjacent H6 helix (R658) form alternate salt bridge interactions with the spindle helix via residue D240. The salt bridges are mutually exclusive. Specifically, in the apo state, R658 establishes a salt bridge

with D240. Upon ATP binding, D240 switches its interaction, forming a salt bridge with R636. We show that the hydrophobic cluster between the spring helix and the Arch domain, and the D240-R636 salt bridge, position the Arch domain in an outward-facing configuration.

### Alternating weak and strong interactions at the opposing ends of the DNA-binding groove enable XPD translocation

During the ATPase cycle, conformational switching of the four XPD domains causes reciprocal DNA binding affinity changes at the opposite ends of the DNA-binding groove. To unravel the origins of these affinity changes, we analyzed persistent XPD-DNA contacts in states S1–S7. Persistent XPD-DNA contacts across all seven macrostates are shown in Fig. 6 and Supplementary Fig. 7. Collectively, contact changes between pairs of consecutive states (S1–S7) define the DNA translocation mechanism of XPD.

XPD's DNA-binding groove accommodates a 12-nucleotide stretch of single-stranded DNA. In the simulations, we identify two well-defined narrow regions along the DNA path, denoted as Constriction 1 and Construction 2 (Supplementary Fig. 8, Supplementary Movie 1). The two constrictions play an integral role in XPD's DNA translocation mechanism. Constriction 1 is positioned at the 5′-end of the ssDNA, while Constriction 2 is located at the 3′-end. Each constriction encloses a stretch of four nucleotides (Supplementary Fig. 8a) while the intervening nucleotides form a bridging segment (Supplementary Fig. 8b). RecA2 encompasses six alpha-helices denoted H1 to H6. Helices H2, H5, H6, and a helical turn (HT) are the structural elements defining Constriction 1. Specifically, the spring helix (H5) and the adjected helix H6, delineate one edge of Constriction 1. Helix H2 and the HT helical turn define the opposite edge (Supplementary Fig. 8c). The narrow cleft between these two helical clusters creates a confined passageway through which ssDNA is threaded. The P-loop is positioned perpendicularly to the spring helix H5 and extends across the path of the ssDNA, effectively marking the entrance to Constriction 1. The rigidity of the helices H2, H5 and H6 and the P-loop imposes a spatial constraint, compelling the ssDNA to traverse the narrow passage in a near-linear conformation.

Embedded within the P-loop, Y625 and Y627 play crucial roles in mediating interactions with the ssDNA at Constriction 1. Specifically, Y627 engages in stacking interactions with the first nucleic acid base (marked in red in Supplementary Fig. 8a and referred to as Marker 1) as the ssDNA enters Constriction 1. On the opposite side of the P-loop, F508 forms additional stacking interactions with the adjacent nucleic acid base within Constriction 1. Concurrently, Y625 positions its side chain parallel to the ssDNA backbone, contributing to the structural alignment of the nucleic acid strand. Besides hydrophobic interactions, positively charged residues (e.g., R511, R683, and R686) line the base of the DNA-binding groove, providing an electrostatically complementary surface to accommodate the ssDNA phosphodiester backbone. Hydrogen-bonding interactions mediated by the polar residues T540, S541, and Q543 provide additional contacts that anchor ssDNA to Constriction 1.

Constriction 2 lies at the intersection of the RecA1, Arch, and Fe–S domains (Supplementary Fig. 8d). A segment from the Fe–S domain (residues 173–175) forms a parallel β-sheet with the core of the RecA1 domain. This β-sheet is the key structural element that optimally orients the Fe–S domain to engage the 3′-end of the DNA. Two α-helices from the Fe–S domain are also prominent in forming Construction 2. One extends parallel to the ssDNA and is denoted as $\alpha_{\parallel}$ (residues 127–137), while the other is oriented perpendicularly and denoted as $\alpha_{\perp}$ (residues 140–147). The $\alpha_{\parallel}$ helix and a helical segment from RecA1 (residues 209–220) define the boundary of Constriction 2. The $\alpha_{\parallel}$ helix and the connector loop to $\alpha_{\perp}$ define the span of Constriction 2 along the ssDNA axis. The Arch domain caps the $\alpha_{\perp}$ helix, encircling ssDNA within the constriction. We identified the following residues as critical for binding and stabilizing ssDNA within the constriction: Fe–S

residues R112, R196, and H135; Arch domain residue H304; and RecA2 residues H210, Y211, I217, and L220. Specifically, H135 from the $\alpha_{\perp}$ helix is strategically positioned at the entrance of Constriction 2 where it engages the first incoming DNA nucleotide denoted as Marker 2 (Supplementary Fig. 8d). At the opposing end, two RecA1 residues H210 and Y211 are poised to stabilize the ssDNA. H210 forms favorable stacking interactions with the exiting nucleotide, while Y211 is aligned to interact with the ssDNA backbone. Positively charged residues (e.g., R112 and R196) that anchor the ssDNA backbone at Constriction 2 are contributed only by the Fe–S domain. In contrast, RecA1 contributes predominantly to hydrophobic or base stacking interactions. Notably, persistent contacts at the two constrictions dominate ssDNA binding while the intervening bridge segment is solvent exposed and maintains only transient contacts with XPD. The one exception is a persistent hydrogen-bonding interaction mediated by R75 from motif Ia (Supplementary Fig. 8b).

DNA translocation through XPD proceeds in two distinct phases. During the S1 → S2 transition, the RecA2 domain moves toward RecA1 while ssDNA slides on the surface of RecA2, advancing through Constriction 1 by a single nucleotide. This shift occurs on a timescale of $3.6 \times 10^{-3}$ s and constitutes the slowest step in the translocation mechanism (Supplementary Fig. 9a). Contact changes during this step are shown in Fig. 6a and Supplementary Fig. 7a. The forward RecA2 displacement enables Marker 1 to navigate past the P-loop, engaging residue F508 of the HT. As a result, a nucleic acid residue from the bridging segment of the ssDNA enters Constriction 1, where it stacks against Y627 of the P-loop. However, the reconfiguration of molecular interactions at Constriction 1 extends beyond the P-loop residues. A concerted exchange of persistent stacking and salt-bridge interactions is observed at Constriction 1, shifting downstream to the neighboring nucleotide. Throughout this process, tight binding to Constriction 2 anchors the 3′-end of the ssDNA.

Following the rotation of RecA2, the H2 helix undergoes a reorientation toward the ssDNA axis (S2 → S3), resulting in narrowing of Constriction 1. This rotation establishes additional DNA contacts, involving residues Y542 of the H2 helix (Supplementary Fig. 7b) and the T628, V626 and V599 residues. Tight binding of the 5′-end of ssDNA is accompanied by a loss of stable interactions at Constriction 2 and a weakened hold on the 3′-end of ssDNA. This modulation of binding affinity sets the stage for the ATP-dependent expansion of Constriction 2 during the S3 → S4 transition (Fig. 6b), eventually leading to the complete disengagement of the Arch and Fe–S domains. This conformational switch represents the second slowest process in the overall mechanism and occurs on a timescale of $4.1 \times 10^{-5}$ s (Supplementary Fig. 9a). The opening of Constriction 2 permits an inward shift of the Fe–S domain toward RecA2 (Fig. 3c), inducing bending of the middle ssDNA bridge segment toward the ATPase cleft.

Upon ATP hydrolysis during the S4 → S5 transition, partial opening of the RecA1–RecA2 interface elicits a backward rotation of the Arch domain. This motion closes Constriction 2 and re-establishes ssDNA binding to the Arch domain (Fig. 6c). In state S5, the Arch domain forms persistent hydrophobic interactions with DNA involving residues A303 and L305 (Fig. 6c). Concurrently, new contacts emerge between the Fe–S domain and DNA that are mediated by residues S111, Y192, and K113. Interestingly, in state S5 Constriction 1 also maintains a moderate hold onto the 5′-end of the ssDNA. Thus, at this stage of the mechanism, both constrictions engage ssDNA to a comparable extent.

The partial opening of the motor domain interface increases the separation between the two constrictions. Since neither constriction releases the DNA, this generates tension on the bridging ssDNA segment, causing it to stretch. The stretched DNA configuration is further stabilized by salt bridge formation with residue K603 of motif V. The resulting tension within the ssDNA bridge induces an inward rotation of both the Arch and Fe–S domains (Fig. 3e). In the subsequent S5 → S6 transition, nucleotide release from the ATPase site triggers additional

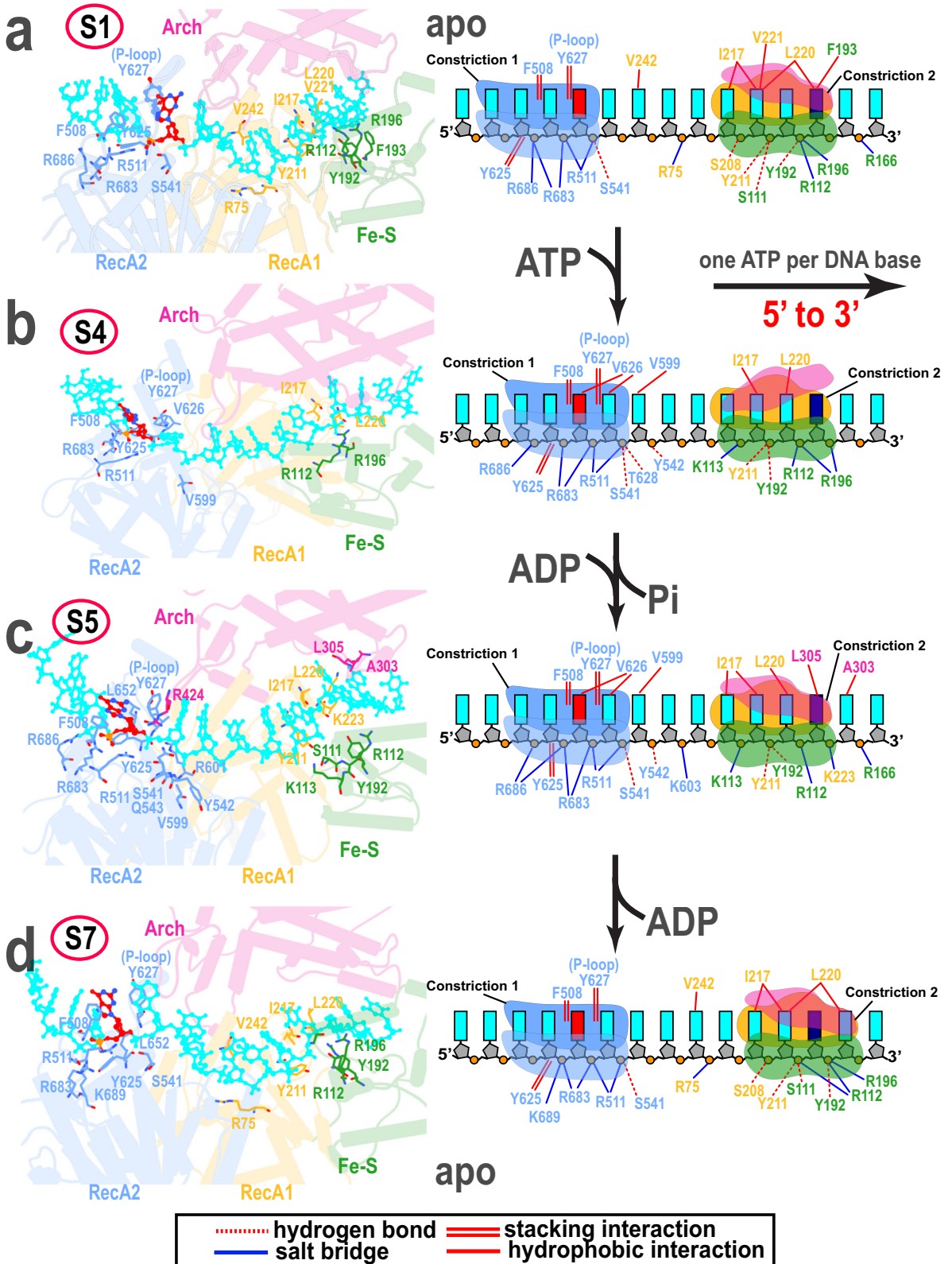

**Fig. 6 | Conformational switching of XPD's domains during the ATPase cycle causes dramatic changes in the ssDNA binding interactions.** Zoomed-in views of the DNA-binding groove of XPD show the evolving contacts between XPD and ssDNA across four functional states: (**a**) S1, (**b**) S4, (**c**) S5, and (**d**) S7. Panels on the left show the DNA binding cleft with bound ssDNA (cyan) and XPD domains colored as follows: RecA1 (gold), RecA2 (blue), Arch (magenta), and Fe–S (green). Key residue contacts to ssDNA are shown explicitly in atomic (ball-and-stick) representation. Side chains of interacting residues are colored by domain. Constriction 1, Constriction 2, and their engagement to ssDNA are shown schematically on the right-side panels, which also depict ssDNA interactions identified by persistent contact analysis of the four macrostates. The reorientation of the Arch domain in the ATP-bound state is shown in panel (**b**).

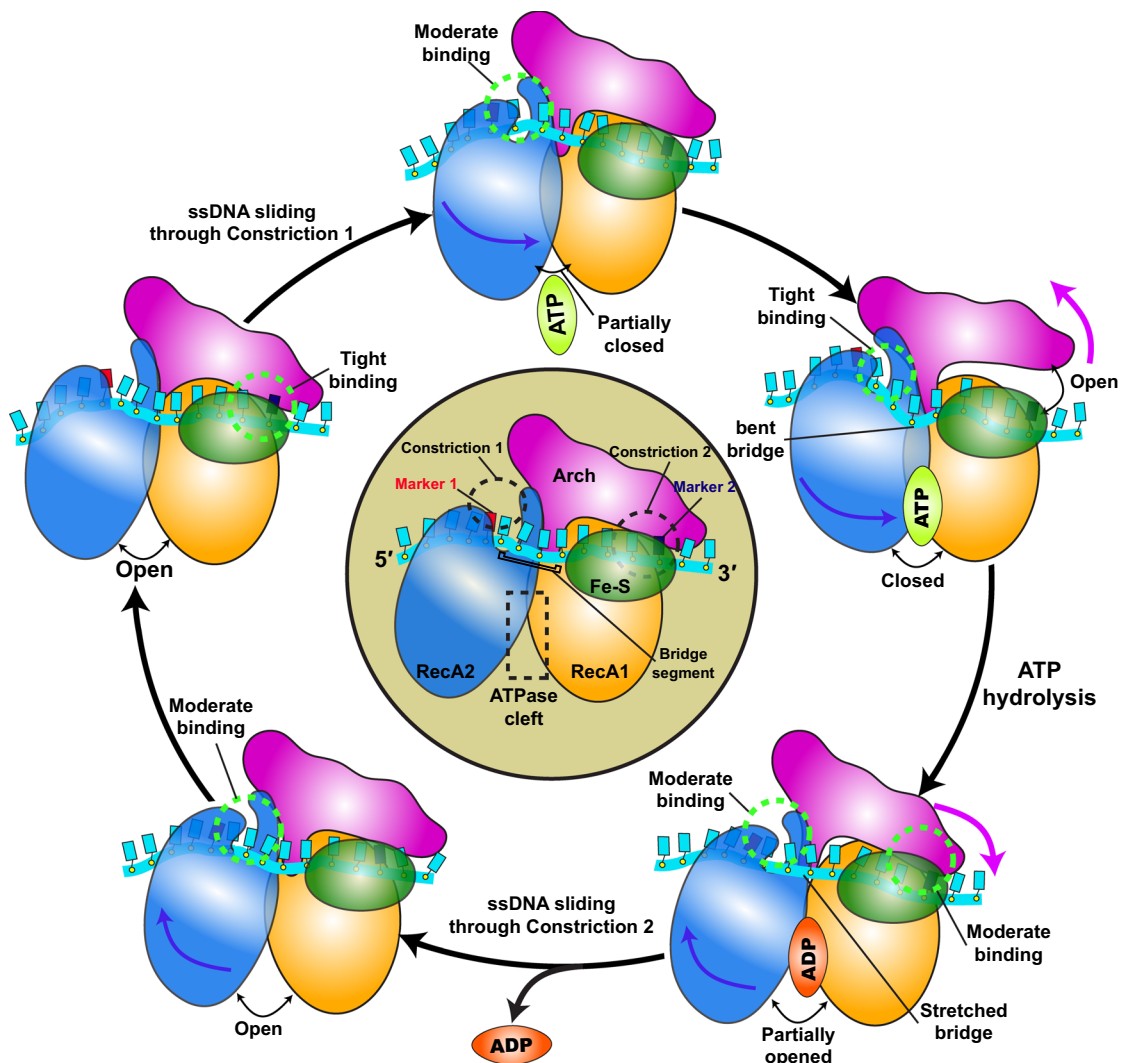

**Fig. 7 | Schematic representation of the XPD translocation mechanism on ssDNA.** The four domains of XPD are color-coded as follows: RecA1 (gold), RecA2 (blue), Arch (magenta), and Fe−S (green). The ssDNA is depicted in cyan. Key structural elements of XPD controlling DNA translocation are shown in the middle of the figure, including Constriction 1, Constriction 2 (black dashed circles) and the ATPase cleft (black dashed rectangle). Marker 1 in red and Marker 2 in navy blue are clearly labeled at the entry point to Constriction 1 and Constriction 2, respectively. Changes in nucleotide state are labeled above the arrows denoting the respective conformational transitions. For each translocation step, the XPD constriction engaged in tightly ssDNA binding is indicated by a light green dashed circle. The directions of the large-scale domain motions are indicated by arrows, each matching the color of the corresponding moving domain.

rotation of RecA2 and releases the moderate hold on ssDNA at Constriction 2. This motion abolishes DNA contacts to Fe−S residue K113, RecA1 residue K223, and Arch domain residues L305 and A303 (Supplementary Fig. 7c). The opening of Constriction 2 relaxes the strain on the middle segment of ssDNA and causes sliding of ssDNA through Constriction 2, thus driving the second phase of DNA translocation during the S6 → S7 transition (Fig. 6d).

While both constrictions are integral to the translocation process, their underlying dynamics are fundamentally distinct. Constriction 1 forms a narrow channel around ssDNA, which remains intact throughout the translocation process. By contrast, Constriction 2 is much more dynamic and capable of opening and closing to accommodate incoming ssDNA. Our analysis reveals reciprocal changes in the number of persistent residue contacts at the two constrictions (Supplementary Fig. 10). Specifically, Constriction 2 establishes maximal contacts with ssDNA in the apo state, where the channel narrows to $6.1 \pm 0.2$ Å, thereby promoting DNA sliding through the more open Constriction 1 ($13.2 \pm 0.2$ Å) during the initial translocation phase. Upon nucleotide exchange at the ATPase site, the system undergoes a dynamic reversal: Constriction 1 increases its affinity for ssDNA,

reaching a maximum in the ATP-bound state with a channel width of $10.7 \pm 0.5$ Å, while Constriction 2 releases the 3′ end by widening to $11.7 \pm 0.8$ Å. This facilitates DNA passage during the subsequent translocation phase. This intricate switch of binding affinities at the two constrictions underpins the 5′–3′ polarity of XPD. A schematic illustrating the overall translocation mechanism is shown in Fig. 7.

### Disease mutations cluster at key positions in XPD that impair DNA binding, translocation or ATP hydrolysis

XPD has essential functional roles in several cellular pathways, including global genome nucleotide excision repair (GG-NER)[3,33–35], transcription-coupled nucleotide excision repair (TC−NER)[36], and transcription initiation[14,36–39]. Predictably, mutations that disrupt XPD's multiple cellular functions cause autosomal recessive genetic disorders—xeroderma pigmentosum (XP), Cockayne syndrome (CS), and trichothiodystrophy (TTD)[19,40–42]. XP mutations are generally associated with GG-NER defects, CS mutations with impaired TC−NER, and TTD with partial transcriptional defects. Combined clinical phenotypes are also possible (e.g., XP/TTD and XP/CS) and are associated with impacts across multiple pathways. To uncover potential links

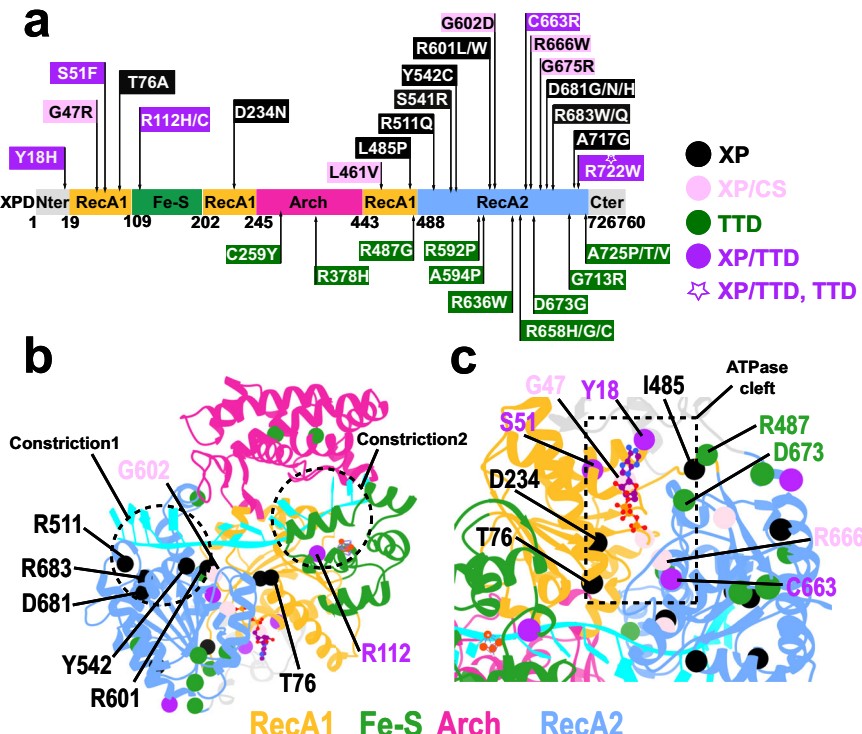

**Fig. 8 | Mapping of XP, CS, and TTD disease mutations onto the XPD structural model. a** Schematic representation of disease-associated mutation sites along the XPD sequence, highlighting mutations linked to xeroderma pigmentosum (XP, black), combined XP and Cockayne syndrome phenotype (XP/CS, pink), trichothiodystrophy (TTD, green), combined XP and TTD phenotype (purple), and overlapping XP/TTD sites (purple star). **b** Positioning of disease mutations in the XPD structural model. **c** Zoomed-in view of mutations lined up along the ATPase cleft of XPD.

between structural and dynamic disruption of XPD's DNA translocation mechanism and disease phenotypes, we analyzed the positioning and effects of XPD missense mutations mapped onto the protein structure (Fig. 8, Supplementary Movie 3). In our analysis, we cluster mutations into three classes: 1) class A involves DNA-binding residues that directly impact DNA translocation; 2) class B mutations interfere with nucleotide binding or impact the opening/closing dynamics of the ATPase cleft; 3) class C mutations involve residues distal from both the DNA-binding groove and ATPase cleft that influence XPD function indirectly via the protein interaction network.

Class A mutations are predominantly associated with the XP phenotype (e.g., R511Q[22], S541R[22], Y542C[22], R601L/W[22], D681G/N/H[22,43] and R683W/Q[22,43,44]), indicating functional impairment of the GG-NER pathway. Other class A mutants feature combined XP/TTD (R112H/C)[22,43,45] phenotype and highlight XPD's involvement in multiple cellular pathways. Class A mutants are irregularly distributed along XPD's DNA-binding groove, clustering mainly within Constriction 1 (R511Q, S541R, Y542C, R601L/W, and R683W/Q). Specifically, residues S541 and Y542 are part of helicase motif IV, while residue R601 is associated with motif V. Located at the base of XPD's ssDNA-binding groove, residues R601 and R683 contact the DNA phosphodiester backbone and provide the necessary electrostatic stabilization to facilitate DNA sliding through Constriction 1. Residue R511 also directly binds and stabilizes ssDNA; however, it is located within the HT helical turn that forms the side wall of the constriction. Mutating these key arginine residues to neutral (e.g., Q) or large hydrophobic ones (L, W) would not only disrupt the favorable electrostatic interactions with ssDNA but also form an obstacle for DNA threading and progression through Constriction 1. Intriguingly, there is a negatively charged residue, D681, situated within Constriction 1 proximal to the DNA-binding groove. Yet, the residue does not interact directly with the groove. Instead, D681 forms a salt bridge with residue R683, helping to optimally position the

arginine with respect to ssDNA. Substitution of D681 with neutral amino acids (e.g., D681G, D681N, or D681H) disrupts this intricate interaction network and impairs DNA translocation through Constriction 1. The S541 and Y542 residues are located upstream of Constriction 1. By forming persistent contacts to ssDNA in states S3, S4 and S5, these residues modulate the ssDNA binding affinity of Constriction 1. Specifically, S541 caps an α-helix from helicase motif IV and aligns the helix dipole to electrostatically favor ssDNA binding. Mutation of S541 to arginine likely destabilizes the helix and disrupts optimal alignment. Likewise, replacement of Y542 with cysteine eliminates the residue's capacity for hydrogen bonding, which is necessary for the transient yet critical contacts enabling the first phase of DNA sliding through Constriction 1. Additionally, a highly conserved residue T76 is positioned at the tip of an α-helix from motif Ia and immediately adjacent to residue R75. While R75 establishes persistent salt-bridge interactions with the bridging segment of the ssDNA, T76 exhibits a more transient mode of engagement. Both residues act in concert to facilitate the second phase of ssDNA sliding. Specifically, R75 engages the phosphodiester backbone, whereas T76 provides additional DNA stabilization through hydrogen bonding. Replacement of T76 with alanine[22] prevents transient electrostatic stabilization of the DNA backbone via R75, disrupting DNA translocation.

The only Class A mutation within Constriction 2 is R112H/C. The R112 residue is highly conserved and positioned on a helical turn near the N-terminus of the Fe−S domain. The residue forms salt bridge interactions with the DNA backbone that facilitate translocation through Constriction 2. Substitution of R112 with histidine or cysteine compromises the Fe−S domain's grip on DNA and likely impairs strand progression through Constriction 2.

Class B mutations include Y18H (XP/TTD)[46], D234N(XP)[22], L461V[44,47–49] and R666W (XP/CS)[22,47–49]. D234 belongs to helicase motif II and, together with E235, coordinates the $Mg^{2+}$ ion that is required for

ATP hydrolysis (Supplementary Fig. 11). Substitution of D234 with a neutral residue (D234N) interferes with magnesium coordination, partially compromises ATP binding, and impairs the ATPase activity. The two XP/CS mutations L461V and R666W affect the integrity of the RecA1–RecA2 interface in the nucleotide-bound states (S4 and S5). R666 is part of the conserved helicase motif VI and plays a crucial role in ATP binding via salt bridge formation to the triphosphate group. Mutation of R666 to tryptophan not only eliminates the positive charge but also inserts a bulky hydrophobic side chain to disrupt the XPD active site. L461 is part of helicase motif II and reinforces the RecA1–RecA2 interface through hydrophobic interactions with residues P694 and W696 of the RecA1 domain (Fig. 4c). L461's hydrophobic side chain forms an essential structural bridge across XPD's motor domain interface. Replacement of L461 with alanine weakens the hydrophobic interactions with the RecA1 domain. Residue Y18 is part of the conserved Q-motif and is positioned at the base of the ATPase cleft. In the apo state, Y18 remains flexible. However, upon nucleotide binding, Y18 stacks against the adenine base to precisely position and stabilize ATP in XPD's active site (Supplementary Fig. 11). Thus, Y18H substitution interferes with optimal ATP binding and hydrolysis.

Class C mutations include G47R (XP/CS)[48,50], and G602D (XP/CS)[22]. Substitution of G47 and G602 affects the dynamics of critically important regions within XPD's DNA binding groove and ATPase cleft, respectively. G47 belongs to a helix from motif I, where its minimal steric footprint provides the necessary spatial clearance for the phosphate backbone of ATP or ADP to be stably accommodated in the active site. Specifically, replacement of G47 with arginine introduces a bulky, positively charged side chain that clashes with the position of the bound nucleotide. In contrast, G602 is located adjacent to the DNA-contacting residue R601 of helicase motif V. The G602 occupies a pivotal junction between a β-sheet and the subsequent helical turn (HT) associated with motif V. The absence of a side chain at this position allows a sharp redirection of the helical turn along the ssDNA axis, creating a narrow channel to accommodate ssDNA sliding. G602D substitution not only introduces a negative charge to disfavor DNA binding but also redirects the protein backbone next to the HT to impair DNA sliding through Constriction 2.

Numerous other XPD residues are involved in disease mutations (R592[22], A594[22], R636[51] and R658[22], D673[22], G675[22], G713[22], A717[47], R722[22], and A725[22]) but do not directly impact the DNA translocation mechanism. Notably, TTD mutations cluster at the interface of XPD with the p44 TFIIH subunit in the transcription pre-initiation complex (PIC)[38]. These mutants impact the structure and dynamics of key interfaces in the PIC and have been annotated previously[23,38].

## Comparative analysis of the XPD and DinG translocation mechanisms

DinG is an XPD homolog from *Escherichia coli* and a DNA helicase with 5′–3′ translocation polarity. XPD and DinG share a common architectural framework, encompassing RecA1, RecA2, Arch, and Fe–S domains (Supplementary Movie 4). The two core motor domains (RecA1 and RecA2) are highly conserved, while the Fe–S and Arch domains show higher sequence divergence. Thus, it is not clear if the two helicases share a common translocation mechanism. To uncover the common and divergent aspects of DNA translocation among SF2 family helicases, we simulated DinG in a way analogous to XPD and compared the translocation mechanisms. Specifically, we modeled the transitions between the apo, ATP-, and ADP-bound states of DinG with the PNEB method employing the same simulation protocol as for XPD (Supplementary Movie 5). Subsequent unbiased MD simulations along the optimal transition path yielded an aggregate sampling of 6 μs per ATPase cycle. MSM analysis of the resulting transition path ensemble identified 9 macrostates, including 5 key on-path metastable intermediates (denoted SD1–SD5) (Supplementary Fig. 12). SD1 and SD5

both represent the apo-DinG state, differing only by the translocation of ssDNA by a single nucleotide. In contrast, states SD3 and SD4 correspond to distinct nucleotide-bound conformations, with SD3 representing the ATP-bound state and SD4 corresponding to the ADP-bound state. To understand the translocation mechanism of DinG in terms of large-scale domain motions, we computed Cα atomic displacements for the transitions among DinG's SD1–SD5 functional states. Porcupine plots showing these conformational rearrangements are presented in Supplementary Fig. 13. During the initial transition (SD1 → SD2), the RecA2 domain rotates toward RecA1 in response to ATP binding, which results in narrowing of the ATPase cleft (Supplementary Fig. 13a). Concurrently, the Fe–S domain shifts toward DinG's ATPase cleft to form an interface with the RecA2 domain. Similar to XPD, the SD1 → SD2 transition constitutes the rate-limiting step in the DinG translocation cycle, with a mean first-passage time of $8.6 \times 10^{-3}$ s (Supplementary Fig. 9b). In the second transition (SD2 → SD3), RecA2 moves further toward RecA1, closing the motor domain interface and optimizing the contacts with ATP (Supplementary Fig. 13b). The closure of the ATPase cleft causes the Arch domain to swing away from the Fe–S domain. During the SD3 → SD4 transition, ATP hydrolysis induces a partial reopening of the ATPase cleft and disrupts the interface between the RecA2 and the Fe–S domains (Supplementary Fig. 13c). In tandem, the Arch domain undergoes a rotation to reestablishing contacts with the Fe–S domain. In the final step, ADP release resets the RecA2 domain to its position in the apo-conformer, which completes the ATPase cycle (Supplementary Fig. 13d).

Persistent contact analysis reveals the gradual contact increase at the RecA1–RecA2 interface during the SD1 → SD2 → SD3 transition (Supplementary Fig. 14a, b). Conversely, interface reopening after ATP hydrolysis results in a marked reduction of interdomain interactions (Supplementary Fig. 14c). A subset of persistent RecA1–RecA2 contacts is retained in the ADP-bound state (SD4), including residues V86, N85, and S459 from RecA1 and residues R569, S593, Q652, I651, and T461 from RecA2. Direct contacts to the ATP or ADP nucleotide are shown in Supplementary Fig. 15. In complete analogy to the XPD mechanism, the DinG spring helix couples the opening/closing dynamics of the motor domains to the movement of the Arch domain. This allosteric coupling is mediated through hydrophobic contacts between P621 of the spring helix and W419 of the Arch domain. The coupling is further reinforced by a salt bridge between E628 of the spring helix and R335 of the Arch domain.

Similar to XPD, the DNA path through DinG traverses two constricted regions—Constriction 1 near the 5′-end and Constriction 2 near the 3′ end. Construction 1 in DinG has notable structural similarity to Construction 1 in XPD. Specifically, Construction 1 in DinG encompasses helices H1, H2, and the intervening helical turn (HT), which define one flank of the constriction; and helices H5 (spring helix) and H6 that form the opposing wall of the construction. Similarity to XPD extends further with the DinG P-loop positioned perpendicularly to the spring helix and demarcating the entry point into Constriction 1. Nonetheless, divergences in the primary sequences of XPD and DinG result in distinct modes of ssDNA engagement by the P-loops of the respective helicases. In XPD, the aromatic residue Y627 engages in DNA base stacking interactions at the entrance of Constriction 1. Equivalent interactions are absent in DinG. Instead, DinG employs a tyrosine distal to the P-loop, Y636, to establish stacking interaction at the 5′-end of the constriction (Supplementary Fig. 16). Overall, the P-loops of both translocases harbor aromatic residues (e.g., F615 of DinG or Y625 of XPD) that stabilize and facilitate DNA passage through the respective constrictions.

By contrast, Constriction 2 in DinG adopts a prominently more open conformation compared to XPD (Supplementary Fig. 17). The pronounced structural divergence stems from differences in the overall architecture of the Fe–S domains of the two helicases. In both translocases, the Fe–S domains are mostly α-helical and positioned

comparably with respect to the motor domains. However, the Fe–S domains differ significantly by fold and primary sequence. In Supplementary Fig. 18a, b, we highlight these differences by presenting a sequence alignment and a structural superposition of the Fe–S domains of XPD and DinG. The XPD Fe–S domain has five α-helices (α1–α5), whereas DinG has six (αD1–αD6). Among these, α5 of XPD and αD6 of DinG show structural correspondence, both harboring a conserved arginine residue (R196 in XPD and R211 in DinG) that is essential for ssDNA binding (Supplementary Fig. 18c, d). Apart from this conserved element, there is low structural similarity between the two Fe–S domains. There are differences in the positioning of the four cysteine residues coordinating the [4Fe–4S] cluster (Supplementary Fig. 18e, f) that affect the conformational flexibility of the domains (Supplementary Fig. 19). In XPD, the cysteine residues coordinating the [4Fe–4S] cluster are distributed throughout the Fe–S domain, whereas in DinG they are positioned near the N- and C-terminal boundaries of the domain. This difference in coordination pattern renders the Fe–S domain of XPD more compact than that of DinG. Furthermore, distinct interdomain interactions arise with the respective RecA1 domains. In XPD, the Fe–S domain is anchored to RecA1 via a parallel β-strand interface (Supplementary Fig. 18g), which constrains Fe–S domain movement during translocation. By contrast, DinG lacks this structural element at the Fe–S/RecA1 interface (Supplementary Fig. 18h), leading to greater flexibility in the Fe–S domain motion (Supplementary Fig. 19).

XPD and DinG also differ in the mode of engagement of the bridging ssDNA segment. In XPD, the DNA is firmly held at the two constrictions while the bridging DNA segment maintains only transient interactions. The only exception is the formation of a persistent salt bridge to the conserved R75 residue. In contrast, DinG engages the bridging segment extensively and forms persistent interactions, including salt bridges with residues R545, K612, and R569, hydrogen bonds with residues Q564 and S593; and hydrophobic contacts mediated by residues I511, I591, and P617 (Supplementary Fig. 20a). Collectively, these interactions anchor ssDNA to the base of the DNA-binding groove and facilitate linear sliding through the DinG groove (rather than DNA bending as observed for XPD). Another difference concerns the contacts formed by the spindle helix. For instance, residue D259 of the spindle helix in DinG forms a hydrogen bond with the nucleotide exiting Constriction 2. A corresponding interaction is absent in XPD.

Similar to XPD, DNA translocation through DinG is a two-stage process. In the initial phase (SD1 → SD3), the ssDNA slides through Constriction 1 as the RecA2 domain moves towards the RecA1 domain. Translocation is coupled to ATP binding and the closure of the ATPase cleft. Persistent ssDNA-protein contacts at Constriction 1 shift downstream by one nucleotide during this phase (Supplementary Fig. 20a, b), while Constriction 2 maintains a tight hold on the 3′ DNA end. RecA2 rotation is accompanied by a directional displacement of the Fe–S domain toward Constriction 1. During this process, we observe the formation of salt-bridge interactions of residues D176 and D173 with R571 of RecA2. The coordinated movement of the Fe–S domain while still maintaining a hold on DNA at Constriction 2 causes sliding of the bridging ssDNA segment toward Constriction 1 in the first phase of translocation. The downstream shift of the bridging segment involves substantial switching of DNA contacts with residues K612, Q564, S593, and R569 (Supplementary Fig. 20a, b). An outward rotation of the Arch domain triggers concurrent opening of Construction 2 associated with partial loss of contacts to the DNA from residues N223, S233, V236 of RecA1, and K191 of Fe–S (Supplementary Fig. 20b). Upon ATP hydrolysis, Constriction 2 recloses, and the ATPase cleft partially opens in state SD4. The complete opening of the ATPase cleft after ADP release in state SD5 triggers the release of the strained DNA conformation between the two constrictions and causes a second stage of sliding ssDNA (Supplementary Fig. 20c, d). The translocation mechanism of DinG is summarized in Supplementary Fig. 21.

## Discussion

XPD is an essential component of the TFIIH assembly, playing a central role in nucleotide excision repair by enabling precise DNA damage verification. Damage verification is driven by XPD conformational switching induced by changing nucleotide state, which allows directional translocation along ssDNA in the 5′-3′ direction. To investigate the molecular basis of this translocation, we integrated existing cryo-EM and crystallographic data with advanced computational modeling to build structural models of XPD in three functional states: apo, ADP-bound, and ATP-bound. Using PNEB simulations, transition path sampling, and Markov state modeling, we mapped the conformational landscape of XPD and identified key intermediate states along its translocation pathway.

Our analysis revealed two discrete constrictions (Constriction 1 and Constriction 2) positioned at the opposite ends of the ssDNA binding groove, which are critical for establishing XPD's 5′-3′ polarity. These constrictions act as molecular clamps, engaging ssDNA in a coordinated manner. Initially, Constriction 2 tightly binds the ssDNA while it slides through Constriction 1. Subsequently, Constriction 1 engages the ssDNA as ssDNA traverses Constriction 2. Notably, the changes in ssDNA binding affinities at the two XPD constrictions occur inversely to those observed in SF1A helicases. Thus, XPD's 5′-3′ polarity is not achieved by inverting the helicase scaffold but by reversing the order of sequential tight binding of the RecA1 and RecA2 domains, consistent with mechanisms observed in SF1B helicases.

Further analysis revealed the dynamic interplay between the XPD motor domains and the two constrictions during ssDNA translocation. The RecA2 domain progressively approaches the RecA1 domain from the apo to the ATP-bound state, reaching its closest proximity in the ATP-bound conformation. In contrast, the Arch domain undergoes a significant rotational shift away from the Fe–S domain in the ATP-bound state while contacting the Fe–S domain in the apo and ADP-bound states. We establish a novel structural relationship between the movement of RecA2 toward RecA1 and the reciprocal rearrangement of the Arch domain, mediated by hydrophobic interactions involving the spring helix of RecA2 and the Arch domain. The coordinated motion of RecA2 and the Arch provides a structural basis for the conformational switching that underpins ssDNA translocation during XPD's ATPase cycle.

Moreover, our results show that DNA- or ATP-binding residues directly engaged in translocation are also hotspots for genetic disease mutations. Specifically, we computed persistent residue contacts across all seven on-path macrostates identified by MSM analysis. We then analyzed the positioning of disease-associated mutations (XP, CS, and combined XP/CS phenotypes) within the XPD protein network for each macrostate to clarify the consequences of XPD functional impairment. Mutations clustered around the ATPase cleft and ssDNA binding groove were categorized into two main groups: 1) mutations that weaken ssDNA binding that are primarily associated with XP, and 2) mutations that rigidify the helicase core or destabilize the RecA1–RecA2 interface—a mechanism principally operational for XP/CS mutations. Among the first class of mutations, we identified several that impair XPD structural elements essential for DNA translocation, providing targets for future functional studies. Disease mutations located outside the ATPase cleft or the DNA binding groove are predominantly associated with the TTD phenotype, suggesting partial transcriptional impairment due to loss of interactions with other subunits of TFIIH or components of the transcription pre-initiation complex.

Additionally, we compare and contrast the translocation mechanisms of XPD and the bacterial homolog DinG, thus shedding light on the common and divergent strategies that SF2 family helicases have evolved to accomplish DNA damage verification in vastly different cellular contexts. Notably, our study significantly extends the previously proposed two-step model for DinG translocation[32]. Our

mechanism retains the essential features of the two-step model: 1) the ATP hydrolysis-induced reciprocal opening/closing of the ATPase cleft and the gap between the Arch and Fe–S domains; and 2) the concomitant alternation in the ssDNA binding affinities of the two motor domains. Yet, we show that two steps are insufficient to fully describe the translocation dynamics. Specifically, by mapping the complete free energy landscape, we identify key metastable intermediate states along the translocation path. This approach yields a more detailed and mechanistically nuanced view of the coordinated motions of the helicase domains and the structural basis for directional ssDNA translocation.

Finally, we comment on the overall DNA translocation timescales (per nucleotide), estimated as ~4 milliseconds for XPD and ~9 milliseconds for DinG. The timescales are underestimated by about an order of magnitude relative to experiments[52,53]. This level of agreement is reasonable when considering the intrinsic limitations of the empirical force field, the accuracy of microstate assignment during MSM construction, and the sampling constraints of MD simulations. It is also worth noting that experimental measurements of XPD kinetics are themselves subject to uncertainties. For instance, Qi et al.[52] reported forward stepping rates of ~10 nucleotides/s (~100 milliseconds/nucleotide) for human XPD obtained from single-molecule optical tweezers experiments. However, helicase activity was monitored under load conditions (i.e., XPD-mediated unwinding of a hairpin DNA substrate). Thus, the unwinding process is not directly comparable to our simulations, which involve only XPD movement on single-stranded DNA but no dsDNA unpairing. Moreover, the overall rate from fitting the unwinding time traces masks the much faster underlying dynamics of hairpin unfolding. The process progresses via repetitive unwinding 'bursts' accompanied by sudden backsliding and re-annealing events, occurring at a rate >100 nucleotides/s (<10 milliseconds/nucleotide). Thus, while fitted kinetic models yield relatively modest forward rates of helicase progression on duplex DNA, the experimentally observed upper bounds of DNA closure indicate that much faster nucleotide-scale motions are involved. Our computed rates, therefore, lie within the experimentally supported dynamic range.

In summary, our findings refine the understanding of XPD's and DinG's functional dynamics during translocation on ssDNA. Importantly, we shed light on the etiology of XPD-associated diseases such as xeroderma pigmentosum and Cockayne syndrome. Collectively, our results provide a foundation for future experimental and theoretical advances to uncover the mechanisms of DNA damage verification in global genome nucleotide excision repair.

## Methods
### Model building
To investigate the translocation mechanisms of XPD and DinG, we constructed models of both helicases bound to single-stranded DNA (ssDNA) in three functional states: apo, ADP-bound, and ATP-bound. The apo XPD structure was derived from the TFIIH–XPA–DNA complex (PDB ID: 6RO4)[2]. Since no human XPD structures in the ATP- or ADP-bound states are currently available, we modeled these nucleotide-bound forms using the *Escherichia coli* DinG structure (PDB ID: 6FWS)[32] as a starting point. Nucleotide binding in the *Escherichia coli* DinG structure induces a conformational rearrangement, characterized by closure of the cleft between the two canonical helicase domains and an opening of the Arch and Fe–S domains. To model a similar ATP-bound conformation in XPD, the RecA1 and RecA2 domains were independently superimposed onto their counterparts in ATP-bound DinG and then assembled to form a closed ATP-binding site. The ATP molecule was positioned based on its location in the DinG–ADP·BeF₃ complex (PDB ID: 6FWS)[32]. While the Fe–S domains of XPD and DinG are composed entirely of α-helices, share a similar topology with respect to Fe–S cluster coordination, and occupy analogous structural positions,

they exhibit notable differences in their fold and primary sequence. To position the Fe-S domain of XPD, the RecA1 and Fe–S domains of apo XPD were aligned as a rigid unit onto the RecA1 domain of DinG. Subsequently, the Arch domain of XPD was placed by superimposing it onto the Arch domain of the DinG–ADP·BeF₃ structure. To construct the ADP-bound model of XPD, the RecA1 domain of apo-XPD was aligned with that of DinG–ADP·BeF₃ structure (PDB ID: 6FWS)[32], and ADP was placed according to its position in the DinG structure. A missing segment in the Arch domain (residues 273–325) of XPD, which was unresolved in the cryo-EM density, was modeled using AlphaFold2 predictions[54,55]. To model DinG in its apo, ADP-bound, and ATP-bound states, we used the apo crystal structure (PDB ID: 6FWR)[32] and DinG–ADP·BeF₃ complex (PDB ID: 6FWS)[32]. The RecA1 domain of apo DinG was aligned with the DinG–ADP·BeF₃ structure (PDB ID: 6FWS)[32], and ADP was positioned according to its coordinates in the DinG structure. The ATP-bound state of DinG was constructed using the DinG–ADP·BeF₃ complex (PDB ID: 6FWS)[32].

### Molecular dynamics simulations
Molecular dynamics (MD) simulations of the XPD and DinG systems were performed on the Summit supercomputer at the Oak Ridge Leadership Computing Facility. System setup was performed using the TLeap module in AMBER18[56]. The TIP3P water model was used to solvate the XPD and DinG systems. To neutralize the net charge, Na⁺ counterions were added, along with extra Na⁺ and Cl⁻ ions to reach a physiological salt concentration of 150 mM. Simulations were run using Amber 18 with the AMBER Parm14SB force field[57] and OL15 nucleic acid modifications[58]. The [4Fe–4S] cluster and its coordinating cysteine residues were modeled using parameters obtained from the AMBER parameter database[59]. Energy minimization was carried out in two stages—150,000 steps using steepest descent, followed by 100,000 steps with the conjugate gradient algorithm. The systems were then heated from 0 to 300 K over 100 ps under NVT conditions, with positional restraints (k = 10 kcal mol⁻¹ Å⁻²) on all heavy atoms of the protein-DNA complex. Equilibration was continued for an additional 5-ns in the NVT ensemble. This was followed by 5 ns of NPT equilibration, gradually releasing the restraints. Production runs were performed in the NPT ensemble (1 atm, 300 K) with a 2-fs integration timestep. Long-range electrostatics were handled with the particle mesh Ewald (PME) method. A 10 Å cutoff was applied to the short-range non-bonded interactions. Bond constraints involving hydrogens were enforced using the SHAKE algorithm. All structural figures were prepared with UCSF Chimera[60].

### Partial nudged elastic band simulations
Conformational transitions among the apo, ATP-bound, and ADP-bound states of XPD and DinG during three successive ATP hydrolysis cycles were modeled using the partial nudged elastic band (PNEB) method[61]. We first extensively equilibrated the apo, ADP-bound, and ATP-bound states with regular unbiased MD. The equilibrated states served as anchor points for initiating PNEB optimizations, which were carried out on a series of replicas of the simulation system starting with the ATP-bound conformer and gradually transitioning to a state where the ATP was swapped out of the active site into bulk solvent and replaced by ADP (from the solvent). Mg²⁺ and phosphate ions were also moved between the active site and solvent to mimic the ATP hydrolysis event. ADP-release from the active site was modeled in the middle of the band and was handled in a similar way by swapping the nucleotide into bulk solvent. For each hydrolysis cycle, 60 replicas were generated to represent intermediate states connecting the ATP, ADP and apo states. PNEB was run with a force constant of 10 kcal mol⁻¹ Å⁻² imposed between neighboring replicas. Initial states were based on the equilibrated apo structures of XPD or DinG. The final states differed only by advancing the ssDNA through the DNA-binding groove by one nucleotide per cycle. Equilibrated ATP- and ADP-bound structures

were used to approximate intermediates. Path optimization was carried out using a simulated annealing protocol: systems were heated from 0 to 300 K over 2 ns with a Langevin thermostat (collision frequency = 1000 ps$^{-1}$), equilibrated at 300 K for 1 ns, and slowly cooled to 0 K over 5 ns. Convergence was assessed by the root-mean-square deviation (RMSD) of backbone atoms, using a threshold of $\Delta$RMSD < 0.3 Å between annealing iterations.

### Transition path ensemble sampling
Unbiased molecular dynamics simulations were initiated from each of the 60 PNEB replicas, each simulated for 100 ns, totaling 6 μs of aggregate sampling per path. Simulations were performed in the NPT ensemble (1 atm, 300 K) with a 2-fs integration timestep using the SHAKE algorithm to constrain bonds involving hydrogens. Simulations were run with PMEMD.CUDA.MPI code (Amber 18) and trajectories analyzed with VMD[62] and UCSF Chimera. Intermolecular interactions were analyzed using the CPPTRAJ module of AMBER18. Hydrogen bonds were defined by a donor–acceptor heavy atom distance of ≤3.2 Å and a donor–hydrogen–acceptor angle of ≥135°. Salt bridges were identified based on a ≤ 3.2 Å distance between oppositely charged residues. π–π stacking interactions were assigned when the distance between the centroids of aromatic rings was ≤3.5 Å. Hydrophobic contacts were determined using a ≤ 4.5 Å cutoff between non-polar atoms. Only interactions that persisted in at least 75% of the trajectory frames were considered stable and included in the final analysis.

### Time-lagged independent component analysis
To analyze the transition path trajectory ensembles in projected collective variable space, we used a dimensionality reduction technique—time-lagged independent component analysis (TICA). TICA was employed using Euclidean distances between Cα atoms for the protein and P atoms for ssDNA to define ten independent components (ICs). Free energy surfaces were projected onto a 2D subspace defined by IC1 and IC2, representing the two slowest collective motions.

### Markov state modeling
Transition path ensembles were clustered into 1500 microstates using k-means clustering. Markov state models (MSMs) were built using the PyEMMA software package[63] with a lag time of 1500 ps, determined by implied timescale analysis. Microstates were grouped into 10 macrostates using the PCCA+ algorithm. Lag time selection and the validation of the MSMs are detailed in Supplementary Fig. 22, Supplementary Fig. 23, respectively. Transition rates between macrostates were estimated as the inverse of the mean first passage time (MFPT).

### Reporting summary
Further information on research design is available in the Nature Portfolio Reporting Summary linked to this article.

## Data availability
Structural models of the XPD complexes corresponding to states S1 through S7 have been deposited in the ModelArchive database with accession codes: 1) ma-rsg0n [https://www.modelarchive.org/doi/10.5452/ma-rsg0n] (Apo state S1); 2) ma-80vsf [https://www.modelarchive.org/doi/10.5452/ma-80vsf] (Intermediate state S2); 3) ma-a80je [https://www.modelarchive.org/doi/10.5452/ma-a80je] (Intermediate state S3); 4) ma-sowmn [https://www.modelarchive.org/doi/10.5452/ma-sowmn] (ATP-bound state S4); 5) ma-6kyfk [https://www.modelarchive.org/doi/10.5452/ma-6kyfk] (ADP-bound state S5); 6) ma-21ute [https://www.modelarchive.org/doi/10.5452/ma-21ute] (Intermediate state S6); 7) ma-d1vqc [https://www.modelarchive.org/doi/10.5452/ma-d1vqc] (Apo state S7). Models of the DinG complexes in states SD1 to SD5 have been deposited in the ModelArchive database with accession codes: 1) ma-mj4dt [https://www.modelarchive.org/doi/10.5452/ma-mj4dt] (Apo state SD1); 2) ma-gcqx8 [https://www.

modelarchive.org/doi/10.5452/ma-gcqx8] (Intermediate SD2); 3) ma-vzytb [https://www.modelarchive.org/doi/10.5452/ma-vzytb] (ATP-bound SD3); 4) ma-wjiwf [https://www.modelarchive.org/doi/10.5452/ma-wjiwf] (ADP-bound state SD4); 5) ma-btvwq [https://www.modelarchive.org/doi/10.5452/ma-btvwq] (Apo state SD5). Initial and final configurations from the MD trajectories of XPD and DinG in nucleotide-free and nucleotide-bound states were included as Supplementary Information. Source data are provided with this paper.

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

## Acknowledgments

We thank Dr. Susan E. Tsutakawa from Lawrence Berkeley National Laboratory for insightful discussions. This work was supported by the National Institute of General Medical Sciences grant R35GM139382, the National Institute of Environmental Health Sciences grant R01 ES032786, the National Science Foundation grant MCB-2027902, and

NCI grants P01 CA092584. An award of computer time was provided by the INCITE program. This research also used resources of the Oak Ridge Leadership Computing Facility, which is a DOE Office of Science User Facility supported under Contract DE-AC05-00OR22725.

## Author contributions

I.I. directed the study. T.P., C.Y., and I.I. contributed to the design of the study. T.P. and C.Y. performed model building and molecular simulations. T.P. C.Y., G.D-B., and I.I. analyzed the data and wrote the manuscript.

## Competing interests

The authors declare no competing interests.
