## [Transparent Peer Review file · Nature Communications]

Translocation mechanism of xeroderma pigmentosum group D protein on single-stranded DNA and genetic disease etiology

Corresponding Author: Professor Ivaylo Ivanov

Version 0:

Reviewer comments:

Reviewer #1

(Remarks to the Author)

The manuscript from Ivanov and colleagues describes extensive computational investigations of the helicase XPD, which is responsible for nucleotide excision repair. Comprising MD simulations, nudged elastic band path optimizations, and various analytical techniques, the calculations reveal the catalytic cycle of helicase activity. Translocation is found to be driven by concerted motions of XPD's domains and specific interactions enabling reptation are identified. These interactions correspond to known hotspots for genetic disease mutations.

This is a well executed study. My questions are mostly about methods and interpretation. In general, there's a lot of quantitative data here but mostly qualitative interpretation. This is not inherently a bad thing, but I think would the paper would benefit from more discussion of the specific numbers determined.

I have a little trouble following the simulation workflow. Since classical MD was used, no ATP hydrolysis steps were actually modeled, nor was product release. How then should I think about a pathway that effectively "jumps over" these steps? Are the various independent simulations run (apo, w/ATP, w/ADP) run and then all the outputs combined for analysis of just the protein and DNA (or just the protein)?

The PMF is presented in Fig. 2a (and Fig. S2), but the discussion on Page 5 is relatively superficial. Are the numbers realistic? What about the barriers between states (or lack thereof)?

It's stated that three of the macrostates identified were taken to be kinetic traps (Page 5). On what basis was this concluded?

It's stated that overall kinetic rates for the translocation processes were determined, but I don't think I see any rates presented anywhere. Am I missing something?

Minor point:

Figure 3d: the Fe-S is partially occluded by panel e.

Reviewer #3

(Remarks to the Author)

Understanding the translocation mechanism by which XPD scans along single-stranded DNA in search of lesions is crucial for comprehending its role in nucleotide excision repair (NER). Yet, its DNA translocation mechanism is incompletely understood. The manuscript authored by Paul et al. is concerned with the mechanism governing translocation polarity in human XPD and its homolog DinG from bacteria. The present work describes the structures and dynamics of XPD and DinG when bound to a ssDNA substrate in apo-, ADP-, and ATP-bound states. The authors have used molecular dynamics simulations together with partial nudged elastic band (PNEB) path optimization, transition path sampling and Markov state modeling (MSM) on these two systems and captured the key intermediate states along the translocation pathways for each system. Along these states, their optimized chain-of-replicas path deciphered the large-scale motions of the XPD's (or DinG's) domains in response to ATP-binding and hydrolysis. Their work provides important insights on both the common

and the diverse mechanistic principles by which XPD and DinG operate during DNA translocation; this sheds light on their distinct functional roles in the eukaryotic and bacteria systems. The work is carefully done and the structural presentations and calculations are of high quality, and thus should advance the field.

Revision is recommended, to address the following issues:

1. The Results Section is lengthy and difficult to follow. The Results could be more focused on the results from computational analyses and could be condensed by removing some of the material. For one example, details of the path optimized translocation procedure should be confined to the Methods Section. Another example, disease mutations and comparison of the XPD and DinG translocation mechanisms are appropriate for the Discussion Section. It is suggested that the authors carefully assess the Results Section and particularly move discussion topics that are interdigitated to the Discussion Section. Additionally, the writing in the Results Section should be sharpened, focused and shortened to emphasize the critical points.
2. In the subsection "Comparative analysis of the XPD and DinG translocation mechanisms", the authors discussed their translocation mechanisms, solely in light of their considerable structural divergences. However, only the functional role of XPD in DNA repair was addressed, while nothing at all was said about the functioning of DinG and its relationship to the translocation mechanism.
3. Page 7, line 190: please change "RecA1-RecA1 interdomain contacts" to "RecA1-RecA2 interdomain contacts"
4. Page 10, lines 303-304: in the sentence "Interestingly, in state S5 Constriction 1 also maintains a moderate hold onto the 5'-end of the ssDNA in S5." Please remove "in S5" at the end of the sentence.
5. Page 11, lines 318-322: please specify the channel widths of Constrictions 1 and 2. Also, please specify the definition of the C<C1> and C<C2> distances in the caption of Supplementary Figure 9.
6. Supplementary Figures 15 and 16 revealed the comparison of the conformations of XPD and DinG at two regions, Constriction 1 near the 5'-end of the ssDNA and Constriction 2 near the 3'-end. However, the corresponding ATPase status isn't mentioned in the manuscript nor in the caption of the figures. Are these illustrated structures extracted from the best representative structures of the specific states (of S1-S7 in XPD and SD1-SD5 in DinG) or taken from the cryo- and/or crystal structures? What is the ATPase status, apo-, ADP-, and ATP-bound states? Please specify.
7. Page 16, lines 488-490: "There are differences in the positioning of the four cysteine residues coordinating the [4Fe-4S] cluster (Supplementary Fig. 17e and 17f) that affect the conformational flexibility of the domains." Please elaborate beyond the single sentence on how the positioning of the four cysteines that coordinate the [4Fe-4S] cluster in DinG contributes to the conformational flexibility of the FeS domain.
8. Page 16, lines 493-494: "By contrast, DinG lacks this structural element at the Fe-S RecA1 interface (Supplementary Fig. 17h), leading to greater flexibility in the Fe-S domain motion (Supplementary Fig. 12)". Are there any MD analyses referring to the greater flexibility of the DinG Fe-S domain compared to XPD?
9. The authors should compare DNA translocation mechanism for DinG of the current work with the previous translocation mechanism of DinG proposed by Cheng et al, 2018 (DNA translocation mechanism of an XPD family helicase. *Elife* 7,e42400).
10. Page 28, line 941: please change "XDP" to "XPD".
11. Do XPD and DinG share the same translocation mechanism on ssDNA as shown in Figure 7, which depicts a schematic representation of the XPD translocation mechanism. If not, a schematic representation for the DinG translocation mechanism should be provided.
12. Supplementary Figure 5: please specify the color codes for the XPD domains.
13. Figure 5a and Supplementary Figure 9a: is the gold domain RecA2 or RecA1?
14. In Figure 6, line 965: "Key residue contacts to ssDNA are shown explicitly in atomic (ball-and-stick) representation. Side chains of interacting residues are colored by domain." However, the left-side panels reveal the side chain of Y627 as green. Does Y627 belong to RecA2, which is colored as blue?
15. The English needs professional editing.

Reviewer #4

(Remarks to the Author)

This is a nice paper that represents a tour-of-force for simulation methods to represent a complex transition. The authors used state of the art methods. I found specially interesting the use of time-lag autoencoder to simplify the transition. I have however many concerns that preclude me to recommend this paper for publication. No clear how metals are treated. For example the Fe-S clusters or bivalent ions, like Zn²⁺ or Mg²⁺ which are known to be

crucial for the biological action of the proteins. It makes little sense to simulate these systems only with NaCl. Under these conditions they are not functional. Details of how these heavy atoms are treated are mandatory, as their absence might contaminate trajectories.

No clear how two components of the autoencoder are able to reproduce the entire transition, for example, is really S2 a cluster, or if looked in one more dimension it will appear as several cluster in this new dimension. Are the barriers and free energy difference consistent with known experimental data.

Not sure the robustness of results with simulation details or simulation length. For example, mapping of clusters on Figure 2 is unclear. It is not clear either the use of Markov State theory here, as MSM samplings are typically obtained in the context of unbiased simulations where thousands of simulations starting from different regions are combined to get a global ensemble of transition probabilities from which guess the entire conformational space.

I like the idea to find explanation to pathological mutations, not a common exercise in biophysical papers. Unfortunately, I do not see how the MD and the dynamics calculations are needed. Most of the mutations could be understood just by looking at the experimental structure of the complex (as it is clear in some of the videos). Calculations showing how dynamics is altered are not provide, and the simplest explanation: mutations affect, structure, or ligand recognition seems valid.

Finally, I am sympathetic with the effort of the authors to write something that is easy to understand looking at the transition in the computer, but difficult to explain. However, the final output is sub_optimal with a very dense and hard to read paper.

Version 1:

Reviewer comments:

Reviewer #1

(Remarks to the Author)

I am happy with the revisions.

Reviewer #3

(Remarks to the Author)

Ivanov and co-workers have done an excellent job in addressing the full set of critiques by all the reviewers, including ourselves, and the manuscript is now acceptable for publication. We are not in agreement with the reviewer who suggested rejection and agree with the Ivanov response to these critiques.

Reviewer #4

(Remarks to the Author)

I think the authors have answered well my criticisms and paper is overall interesting. I think it is worth to publish

Dear Editor,

We thank you and the three reviewers for their careful consideration of the manuscript and their positive feedback.

The reviewers noted several suggested improvements that we strived to accomplish in the revised manuscript. Our responses are in blue font here and in the revised manuscript.

Point by point response to reviewer comments:

Reviewer #1 (Remarks to the Author):

The manuscript from Ivanov and colleagues describes extensive computational investigations of the helicase XPD, which is responsible for nucleotide excision repair. Comprising MD simulations, nudged elastic band path optimizations, and various analytical techniques, the calculations reveal the catalytic cycle of helicase activity. Translocation is found to be driven by concerted motions of XPD's domains and specific interactions enabling reptation are identified. These interactions correspond to known hotspots for genetic disease mutations.

This is a well-executed study. My questions are mostly about methods and interpretation. In general, there's a lot of quantitative data here but mostly qualitative interpretation. This is not inherently a bad thing, but I think the paper would benefit from more discussion of the specific numbers determined.

i) I have a little trouble following the simulation workflow. Since classical MD was used, no ATP hydrolysis steps were actually modeled, nor was product release. How then should I think about a pathway that effectively "jumps over" these steps? Are the various independent simulations run (apo, w/ATP, w/ADP) run and then all the outputs combined for analysis of just the protein and DNA (or just the protein)?

We thank the reviewer for pointing out the need to further clarify the details of our path optimization protocol. The reviewer is completely correct that classical MD simulations cannot explicitly handle chemical transformations such as ATP hydrolysis. Yet, it is possible to use classical MD to probe conformational transitions induced by switching the nucleotide state of the ATPase. In the PNEB simulations this is accomplished by exchanging the nucleotides (e.g., ATP) between the active site and bulk solvent. This exchange is necessitated by the requirement of the PNEB method to have an identical number of atoms in each replica of the simulated system. We first extensively equilibrate the apo, ADP-bound and ATP-bound states with regular unbiased MD. The equilibrated states serve as anchor points for initiating PNEB optimization, which is carried out on a series of replicas of the simulation system starting with the ATP-bound conformer and gradually transitioning to a state where the ATP has been swapped out of the active site into bulk solvent and replaced by ADP (from the solvent). Mg and phosphate ions are also moved between the active site and solvent to mimic the ATP hydrolysis event, which is itself abrupt but followed by more gradual conformational changes. ADP-release from the active site also occurs in the middle of the band and is handled in a similar way by swapping the nucleotide into bulk solvent. Once the PNEB path optimization is converged, all replicas are subjected to unbiased MD simulation to sample the conformational space along the optimal path. The unbiased MD trajectories are combined prior to MSM analysis and thus the MD ensemble features a mixture of conformers originating from

different replicas that are either apo, ATP- or ADP-bound. These conformers seamlessly interpolate between the three functional states to represent the ATPase cycle. We have revised the Methods section of the manuscript to better explain how the simulations handle ATP hydrolysis and product release.

ii) The PMF is presented in Fig. 2a (and Fig. S2), but the discussion on Page 5 is relatively superficial. Are the numbers realistic? What about the barriers between states (or lack thereof)?

The free energy landscape presented in Fig. 2a was computed by projecting the MD ensemble from a high dimensional space into the space of only two ICs. The result of this over-projection is the lowering of the apparent barriers between macrostates, which is entirely expected. The MSM macrostate transition rates are not computed directly from the barriers but from the analysis of microstate transition probabilities. Nonetheless, the computed overall timescale for DNA translocation (~ 4 milliseconds/nt) is underestimated by an order of magnitude compared to the experimental timescale. The agreement is quite reasonable given the expected accuracy of the empirical force field, the accuracy of microstate assignment during MSM construction, and the sampling limitations of MD. We note that the experimental timescale for XPD is also subject to uncertainties/limitations. For instance, Qi et al. (Qi, Z., *et al.* eLife, 2013, doi.org/10.7554/eLife.00334) reported forward stepping rates of ~ 10 nucleotides/s (~ 100 milliseconds/nucleotide) for human XPD obtained from single-molecule optical tweezers experiments. However, helicase activity was monitored under load conditions (i.e., XPD-mediated unwinding of a hairpin DNA substrate). Thus, the unwinding process is not directly comparable to our simulations, which involve only XPD movement on single-stranded DNA but no dsDNA unpairing. Moreover, the overall rate from fitting the unwinding time traces masks the much faster underlying dynamics of hairpin unfolding. The process progresses via repetitive unwinding ‘bursts’ accompanied by sudden backsliding and re-annealing events, occurring at a rate >100 nucleotides/s (<10 milliseconds/nucleotide). Thus, while fitted kinetic models yield relatively modest forward rates of helicase progression on duplex DNA, the experimentally observed upper bounds of DNA closure indicate that much faster nucleotide-scale motions are involved. Our computed rates therefore lie within the experimentally supported dynamic range. We have now included a separate section in the Discussion specifically addressing this point.

iii) It's stated that three of the macrostates identified were taken to be kinetic traps (Page 5). On what basis was this concluded?

The minimum free energy path (MFEP) determined by PNEB was projected onto the space defined by the first two independent components IC1 and IC2. Macrostates that were not traversed by the MFEP were denoted as kinetic traps.

Figure (for review only). Projection of the PNEB path (black line) onto the free energy landscape of XPD translocation. Macrostates identified as kinetic traps are indicated by red lines.

iv) It's stated that overall kinetic rates for the translocation processes were determined, but I don't think I see any rates presented anywhere. Am I missing something?

Thank you for pointing this out. We have now provided a new supplementary figure (Supplementary Figure 9), which shows the computed timescales (i.e., the inverse rates) for forward and backward transitions between all pairs of MSM macrostates. These were obtained from mean first passage time analysis of the MSMs for a) XPD and b) DinG helicase, respectively. The timescales are shown as graphs wherein vertices represent the MSM macrostates and edges (arrows) represent the forward and backward transition timescales.

Supplementary Figure 9. Forward and backward transition timescales between macrostates from mean first passage time analysis for a) XPD and b) DinG. Vertices denote MSM macrostates. Directed edges (black arrows) represent forward and backward transitions between pairs of macrostates. Edges are labelled with the corresponding timescale values.

Other points for revising the manuscript:

1) Figure 3d: the Fe-S is partially occluded by panel e.

Thank you for pointing this out. This issue is now corrected in the revised Figure 3.

Reviewer #3 (Remarks to the Author):

Understanding the translocation mechanism by which XPD scans along single-stranded DNA in search of lesions is crucial for comprehending its role in nucleotide excision repair (NER). Yet, its DNA translocation mechanism is incompletely understood. The manuscript authored by Paul et al. is concerned with the mechanism governing translocation polarity in human XPD and its homolog DinG from bacteria. The present work describes the structures and dynamics of XPD and DinG when bound to a ssDNA substrate in apo-, ADP-, and ATP-bound states. The authors have used molecular dynamics simulations together with partial nudged elastic band (PNEB) path optimization, transition path sampling and Markov state modeling (MSM) on these two systems and captured the key intermediate states along the translocation pathways for each system. Along these states, their optimized chain-of-replicas path deciphered the large-scale motions of the XPD's (or DinG's) domains in response to ATP-binding and hydrolysis. Their work provides important insights on both the common and the diverse mechanistic principles by which XPD and DinG operate during DNA translocation; this sheds light on their distinct functional roles in the eukaryotic and bacteria systems. The work is carefully done and the structural presentations and calculations are of high quality, and thus should advance the field.

Revision is recommended, to address the following issues:

1. The Results Section is lengthy and difficult to follow. The Results could be more focused on the results from computational analyses and could be condensed by removing some of the material. For one example, details of the path optimized translocation procedure should be confined to the Methods Section. Another example, disease mutations and comparison of the XPD and DinG translocation mechanisms are appropriate for the Discussion Section. It is suggested that the authors carefully assess the Results Section and particularly move discussion topics that are interdigitated to the Discussion Section. Additionally, the writing in the Results Section should be sharpened, focused and shortened to emphasize the critical points.

We thank the reviewer for the suggestion to streamline the Results section. We have implemented some of the recommended changes by moving simulation protocol details to the Methods section. However, leaving the section on disease mutations separate from the Discussion section was a conscious decision on our part and we have left it unchanged. Our rationale is that the paper needs to serve multiple audiences – computationally inclined readers and/or structural biologists will likely focus on the MSM results and the discussion of structural changes that drive the XPD or DinG translocation mechanisms. By contrast, experimental biochemists, geneticists, or clinicians may focus on the disease mutant section. Indeed, it may be the only section they read from the manuscript. Thus, we felt it was important to keep this section distinct.

2. In the subsection “Comparative analysis of the XPD and DinG translocation mechanisms”, the authors discussed their translocation mechanisms, solely in light of their considerable structural divergences. However, only the functional role of XPD in DNA repair was addressed, while nothing at all was said about the functioning of DinG and its relationship to the translocation mechanism.

As noted by the reviewer, the emphasis of the paper is clearly on the mechanism of human XPD. This choice is justified by the critical significance of XPD for the assembly and function of the human nucleotide excision repair machinery. In this respect, the activity and translocation mechanism of XPD carry outsized importance for understanding the link between NER impairment and human disease etiology. To place XPD's mechanism in a broader context, we also performed comparative analysis with XPD's bacterial homolog DinG. While both enzymes share key structural features, they employ distinct strategies for ssDNA engagement and translocation. This comparison underscores both the evolutionary conservation and divergence of SF2 helicases and is relevant for understanding the fundamental mechanisms of genome maintenance across species. Nonetheless, the DinG mechanism does not carry the same significance for understanding human disease and was consciously deemphasized due to limitations of space. Indeed, many of the figures relevant to DinG were placed in Supplementary Information. We note that there was one omission related to DinG that we have corrected in revision, and we thank the reviewer for bringing it to our attention. We have now included a schematic representation of DinG's translocation mechanism in Supplementary Fig. 21. This figure is the counterpart of Figure 7 in the main text and will hopefully give the readers a more direct comparison of the two helicase mechanisms.

Other points for revising the manuscript:

1. *Page 7, line 190: please change "RecA1-RecA1 interdomain contacts" to "RecA1-RecA2 interdomain contacts"*

Thank you. The typo was corrected.

2. *Page 10, lines 303-304: in the sentence "Interestingly, in state S5 Constriction 1 also maintains a moderate hold onto the 5'-end of the ssDNA in S5." Please remove "in S5" at the end of the sentence.*

Done.

3. *Page 11, lines 318-322: please specify the channel widths of Constrictions 1 and 2. Also, please specify the definition of the $C<C1>$ and $C<C2>$ distances in the caption of Supplementary Figure 9.*

We have now provided ranges (in Angstroms) for the channel widths of Construction 1 and Constriction 2 in the text of the manuscript. We have also included the definitions of the C_{C1} and C_{C2} distances in the caption of revised Supplementary Figure 10.

4. *Supplementary Figures 15 and 16 revealed the comparison of the conformations of XPD and DinG at two regions, Constriction 1 near the 5'-end of the ssDNA and Constriction 2 near the 3'-end. However, the corresponding ATPase status isn't mentioned in the manuscript nor in the caption of the figures. Are these illustrated structures extracted from the best representative structures of the specific states (of S1-S7 in XPD and SD1-SD5 in DinG) or taken from the cryo- and/or crystal structures? What is the ATPase status, apo-, ADP-, and ATP-bound states? Please specify.*

Comparison XPD and DinG in the original Supplementary Figures 15 and 16 was done using their apo structures. This is now explicitly stated in the figure captions of revised Supplementary Figures 16 and 17.

5. Page 16, lines 488-490: “There are differences in the positioning of the four cysteine residues coordinating the [4Fe-4S] cluster (Supplementary Fig. 17e and 17f) that affect the conformational flexibility of the domains.” Please elaborate beyond the single sentence on how the positioning of the four cysteines that coordinate the [4Fe-4S] cluster in DinG contributes to the conformational flexibility of the FeS domain.

We have incorporated the following section into the revised manuscript:

“In XPD, the cysteine residues coordinating the [4Fe-4S] cluster are distributed throughout the Fe-S domain, whereas in DinG they are positioned near the N- and C-terminal boundaries of the domain. This difference in coordination pattern renders the Fe-S domain of XPD more compact than that of DinG.”

6. Page 16, lines 493-494: “By contrast, DinG lacks this structural element at the Fe-S RecA1 interface (Supplementary Fig. 17h), leading to greater flexibility in the Fe-S domain motion (Supplementary Fig. 12)”. Are there any MD analyses referring to the greater flexibility of the DinG Fe-S domain compared to XPD?

Thank you for pointing this out. We now present computed B-factors mapped onto apo-XPD and apo-DinG. The Fe-S domains are outlined by black dash-line circles. The new supplementary figure clearly shows that the apo-DinG Fe-S domain exhibits greater flexibility and is more dynamic than the corresponding XPD Fe-S domain.

Supplementary Figure 19. Computed B-factors mapped onto the structures of (a) apo-XPD and (b) apo-DinG. The Fe-S domains are outlined by black dashed-line circles.

7. The authors should compare DNA translocation mechanism for DinG of the current work with the previous translocation mechanism of DinG proposed by Cheng et al, 2018 (DNA translocation mechanism of an XPD family helicase. *Elife* 7,e42400).

Our proposed mechanism for DinG is an elaboration on the previous two-step translocation mechanism set forth in Cheng et al, *Elife* (2018). Naturally, our mechanism encompasses more

discrete states and includes more steps (Supplementary Fig. 21). However, the overall domain motions that drive translocation are consistent between the two models. This is now explicitly stated in the Discussion section:

“Notably, our study significantly extends the previously proposed two-step model for DinG translocation³². Our mechanism retains the essential features of the two-step model: 1) the ATP hydrolysis induced reciprocal opening/closing of the ATPase cleft and the gap between the Arch and Fe-S domains; and 2) the concomitant alternation in the ssDNA binding affinities of the two motor domains. Yet, we show that two steps are insufficient to fully describe the translocation dynamics. Specifically, by mapping the complete free energy landscape, we identify key metastable intermediate states along the translocation path. This approach yields a more detailed and mechanistically nuanced view of the coordinated motions of the helicase domains and the structural basis for directional ssDNA translocation.”

8. Page 28, line 941: please change “XDP” to “XPD”.

The typo has been corrected.

9. Do XPD and DinG share the same translocation mechanism on ssDNA as shown in Figure 7, which depicts a schematic representation of the XPD translocation mechanism. If not, a schematic representation for the DinG translocation mechanism should be provided.

The XPD and DinG mechanisms are consistent with respect to the overall domain rearrangements that drive translocation. However, key differences exist in the role and flexibility of the Fe-S domains and in the way single-stranded DNA is accommodated by the XPD and DinG DNA-binding grooves. To better emphasize these distinctions, we have now included a schematic representation of the DinG translocation mechanism as new Supplementary Figure 21.

10. Supplementary Figure 5: please specify the color codes for the XPD domains.

Color codes for the XPD domains were added to the Supplementary Figure 5 caption.

11. Figure 5a and Supplementary Figure 9a: is the gold domain RecA2 or RecA1?

The domain in gold is RecA1. This has been corrected in new Figure 5a and new Supplementary Figure 10a.

12. In Figure 6, line 965: “Key residue contacts to ssDNA are shown explicitly in atomic (ball-and-stick) representation. Side chains of interacting residues are colored by domain.” However, the left-side panels reveal the side chain of Y627 as green. Does Y627 belong to RecA2, which is colored as blue?

We thank the reviewer for pointing this out. Y627 is part of the RecA2 domain and should therefore be shown in blue. This has been corrected in Figure 6 of the revised manuscript.

13. The English needs professional editing.

We have made an effort to streamline the Results section and have corrected some typos that escaped our attention during the initial submission. We do not think additional language editing is needed.

Reviewer #4 (Remarks to the Author):

This is a nice paper that represents a tour-of-force for simulation methods to represent a complex transition. The authors used state of the art methods. I found specially interesting the use of time-lag autoencoder to simplify the transition.

I have however many concerns that preclude me to recommend this paper for publication.

1) No clear how metals are treated. For example the Fe–S clusters or bivalent ions, like Zn^{2+} or Mg^{2+} which are known to be crucial for the biological action of the proteins. It makes little sense to simulate these systems only with NaCl. Under these conditions they are not functional. Details of how these heavy atoms are treated are mandatory, as their absence might contaminate trajectories.

We thank the reviewer for raising this point. Indeed, our simulation systems include an iron sulfur cluster [4Fe-4S] and a Mg^{2+} ion in the ATPase active site. However, it does not include Zn^{2+} as the XPD protein does not contain any zinc binding motifs (e.g., no zinc fingers). The Fe–S cluster plays a purely structural role in maintaining the structural integrity of the domain in which it resides. The Mg ion is essential for ATP hydrolysis by the XPD and DinG helicases and we have now explained more clearly how it is treated in the PNEB simulations (see response to Reviewer #1). We have added to the Methods section a more detailed description of how the [4Fe-4S] cluster and Mg^{2+} ions were modeled and treated in the simulations.

2) No clear how two components of the autoencoder are able to reproduce the entire transition, for example, is really S2 a cluster, or if looked in one more dimension it will appear as several cluster in this new dimension. Are the barriers and free energy difference consistent with known experimental data.

This is a common concern when working in projected space and we have given it due attention. Specifically, we plotted projections onto subspaces defined by other ICs. For example, in the figure below we show the projection onto the second and third time-lagged independent components (IC2 versus IC3). Additionally, we map 1000 frames selected from macrostates S1–S7 onto the new free energy landscape defined by IC2 & IC3. We verify that the same sequence of metastable states (S1–S7) is observed on the IC2 vs IC3 plot and the path connecting them is clearly identifiable and traverses them (see figure below).

We also point out that working in subspaces of more than two dimensions is possible but makes the results difficult to visualize. For instance, it requires taking multiple slices from 3- or 4-dimensional PMFs. Moreover, not being able to properly visualize the PMF in high-dimensional space carries the risk of not noticing sampling discontinuities along the optimal path.

Regarding the computed PMF barriers and comparison to the experimental XPD translocation rate please refer to our response to the second question of Reviewer #1.

Figure (for review only). Projection of the free energy landscape for XPD onto the 2D subspace defined by IC2 and IC3. A total of 1000 frames from each original microstate (S1–S7) were projected onto this space. Macrostates are color-coded and labeled consistently with Figure 2 in the main text.

3) *Not sure the robustness of results with simulation details or simulation length. For example, mapping of clusters on Figure 2 is unclear. It is not clear either the use of Markov State theory here, as MSM samplings are typically obtained in the context of unbiased simulations where thousands of simulations starting from different regions are combined to get a global ensemble of transition probabilities from which guess the entire conformational space.*

Our computational strategy is not any different from what the reviewer described. Instead of thousand short simulations, we employ hundreds of somewhat longer (100-ns) unbiased MD runs. These are initiated from the PNEB replicas along the optimized path and, therefore, represent different regions of phase space. Importantly, only unbiased simulation trajectory data is used in constructing the MSM models. By choosing the PNEB replicas as starting points we simply focus the unbiased sampling on the most relevant regions of phase space and avoid sampling regions far away from the MFEP path that are irrelevant for XPD translocation. Finally, our aggregate sampling is very extensive (6 μ s/path x 3 independent paths = 18 μ s for XPD and the same sampling for DinG). This compares favorably to the current state of the art in MSM modeling.

Regarding the robustness of our conclusions, we note that the MSM results were successfully replicated in three independently simulated consecutive translocation cycles. The corresponding free energy landscapes are provided in Supplementary Figure 3.

4) *I like the idea to find explanation to pathological mutations, not a common exercise in biophysical papers. Unfortunately, I do not see how the MD and the dynamics calculations are needed.*

Most of the mutations could be understood just by looking at the experimental structure of the complex (as it is clear in some of the videos). Calculations showing how dynamics is altered are not provide, and the simplest explanation: mutations affect, structure, or ligand recognition seems valid.

We respectfully disagree with the reviewer's assessment. First, the effects of disease mutations have to be evaluated in the context of XPD's ATPase and DNA translocation mechanisms. This entails knowledge of all functionally relevant states along the ATPase cycle. Such knowledge had not been available for XPD prior to our paper, either from experiments or from simulation. For instance, no experimental structure of ATP-bound human XPD is available. Without the MD simulations, path optimization and MSM classification of the intermediates along XPD's translocation cycle one could not provide a confident interpretation of the effects of disease mutations even at the level of structure or ligand recognition. Furthermore, we analyze the global motions of XPD and DinG's domains along each step in the respective mechanisms (see porcupine plots) and discuss disease mutations in the context of these dynamic rearrangements.

We agree with the reviewer that the impact of the mutations is primarily on the local protein structure, domain interface stability, nucleotide recognition in the ATPase active site or ATP hydrolysis. Distal allosteric effects play a secondary role.

Finally, I am sympathetic with the effort of the authors to write something that is easy to understand looking at the transition in the computer, but difficult to explain. However, the final output is sub_optimal with a very dense and hard to read paper.

We agree that the paper is relatively long and covers a lot of ground. In part, our difficulty has been that we attempted to combine material worthy of two separate manuscripts into one and make it as comprehensive as possible. Thus, we cover not only the XPD translocation mechanism but also the bacterial counterpart DinG mechanism, and the impact of disease mutations on XPD function. On the flip side, there are aspects of the manuscript that would make it appealing to multiple audiences – computational biologists, structural biologists, biochemists, geneticists and clinicians. Given the extensive results, we do not think the paper could be shortened considerably without affecting its impact.

We appreciate the reviewers' efforts in considering our revised manuscript. Thank you for considering our paper for Nature Communications.

Sincerely,

Ivaylo Ivanov